



# Relative importance of uncertain model parameters driving water fluxes in a Land Surface Model

David Luttenauer[1], Aronne Dell'Oca[2], Alberto Guadagnini[2], Sylvain Weill[1], Philippe Ackerer[1]

[1] Institut Terre et Environnement de Strasbourg, Université de Strasbourg, CNRS, ENGEES, F-67000 STRASBOURG, France

[2] Dipartimento di Ingegneria Civile e Ambientale, Politecnico di Milano, Milano, Italy

Corresponding author: Philippe Ackerer, ackerer@unistra.fr

## Abstract

We focus on the way temporal distributions of key components of the water cycle are influenced by typically uncertain parameters embedded in a Land Surface Model. We rest on a joint analysis of multiple global sensitivity metrics to provide a comprehensive assessment of the ranking of the relative importance of uncertain factors of various origins on the hydrological system response. The latter is rendered in terms of the temporal dynamics of transpiration, evaporation, and groundwater recharge. The NIHM (Normally Integrated Hydrological Model) modular Land Surface model is applied to simulate realistic field conditions (in terms of, e.g., climate, vegetation, and soil type) associated with two watersheds in the Vosges region (France) across a one-year period. These watersheds are characterized by similar climatic conditions while being associated with differing soil types and vegetation. Uncertain model parameters we consider comprise monthly values of albedo and leaf area index, vegetation-related parameters, as well as parameters related to the soil types associated with the litter layer and root zone. Four diverse sensitivity indices are used to quantify impacts of uncertain model parameters on the whole probability distribution or given statistical moments of the density





function of model outputs. Our results document that the strength of the relative importance of model
parameters depends on the statistical moment considered. Evaporation is directly influenced by the
energy flow through the canopy and by the parameters associated with the top litter layer. As one could
expect, transpiration appears as mainly influenced by the vegetation characteristics and by albedo that
influences the incoming radiation. Groundwater recharge is influenced only by a very limited number
of model parameters. It mainly depends on soil-related parameters and is unexpectedly not sensible to
any of the vegetation parameters considered, except the root layer thickness and the intercept.





## 1 Introduction

Since the work of Manabe (1969), Land Surface Models (LSMs) have become critical tools for modeling energy balance, water cycle, vegetation dynamics, and their feedbacks. They constitute one of the key routines employed in General Circulation Models (GCMs) to evaluate the effects of climate change on the Earth surface as well as in modeling workflows routinely used for water resources management. Numerous LSMs have been developed in the last decades (e.g. Blyth et al., 2020; Fisher and Koven, 2020; Overgaard et al., 2006; and references therein). These are characterized by various levels of complexity such as, e.g., LSMs described in Niu et al., (2011), Maneta and Silverman (2013), Decharme et al., (2019), Lawrence et al., (2019), Wiltshire et al., (2019) or Yokohata et al. (2019).

A primary purpose of a LSM is to simulate exchanges of energy and water between the land surface, the underground, and the atmosphere. Due to the variety of processes that are mathematically rendered therein, LSMs embed numerous input parameters related to vegetation, energy transfer and water fluxes across the atmosphere and in the soil. Many of these parameters (e.g., soil attributes or vegetation characteristics) are difficult to quantify through direct measurements and may vary across scales and locations. These elements typically lead to uncertainty in our knowledge of the values of such parameters (Beven and Smith, 2014). In case a target model output is not (or only minimally) affected by the particular value associated with certain parameters, it may be appropriate to rely on typical literature values for these. It is then important to properly identify parameters that significantly impact model outputs.

In this context, sensitivity analysis enables one to quantitatively rank the influence that diverse uncertain model parameters, involved in the mathematical rendering of different processes, have onto model predictions of interest. Thus, sensitivity analysis should be considered as an integral and essential step in the diagnosis and understanding of complex models of hydrological systems (Ferretti et al., 2016; Song et al., 2015; Vemuri et al., 1969; Razavi et al., 2021). Sensitivity analyses can yield



valuable information about LSMs development, potentially simplifying mathematical representations,
and streamline LSMs calibration by omitting uninfluential parameters (Mc Cuen, 1973).
Demarty et al. (2005) perform a sensitivity analysis of soil heat conduction flux, sensible heat flux,
latent heat flux, water content of the upper five soil centimeters and local directional brightness
temperature considering 35 input parameters associated with the physically-based model SiSPAT-RS
(Braud et al., 1995). Their findings indicate the saturated water content of the upper 5 cm of soil and
the thermal infrared brightness temperature as the most influent model parameters. Liang and Guo
(2003) compare the sensitivity of evapotranspiration, total runoff, sensible heat flux and soil moisture
to 5 model parameters that appear in 10 different LSMs. Their results show that parameters associated
with soil properties appear to play a more significant role than those associated with vegetation
properties whereas the outputs of the diverse models considered exhibit the highest sensitivity to the
maximum soil moisture content, considering three different hydroclimatic scenarios. Bastidas et al.
(2006) assess parameter sensitivity of 5 different LSMs with increasing level of complexity in the
description of vegetation-related physical processes. They show that (*a*) the sensitivity of the energy
budget component to parameters with similar physical meaning employed in the diverse LSMs
analyzed depends on the specific LSM model and varies depending on the location of the system, and
(*b*) soil-related parameters could be considered as most influential. Based on the hydrologic model
WetSpa (Wang et al., 1996), Yang et al. (2012) highlight the intense sensitivity of runoff flow rate of
two water catchments to the parameters involved in the description of the evapotranspiration process.
Li et al. (2013) employ diverse sensitivity analysis methodologies to assess the sensitivity of 6 model
outputs of the LSM CoLM (Dai et al., 2003), i.e., sensible heat, latent heat, upward longwave radiation,
net radiation, soil temperature, and soil moisture, with respect to 40 uncertain model parameters. Their
results highlight that all model outputs are sensitive to the Clapp and Hornberger parameter (which is
related to soil water retention, see Clapp and Hornberger, 1978), while (*i*) aerodynamic roughness
length markedly influences the sensible and latent heat fluxes (along with the upward longwave and





net radiations and soil temperature), and (*ii*) soil porosity chiefly governs soil moisture. Li et al. (2013)
suggest that latent heat flux (related to evapotranspiration) is also sensitive to quantum efficiency of
vegetation photosynthesis and minimum soil suction. Baroni and Tarantola (2014) employ classical
variance-based Sobol indices to rank the importance of model parameters and forcing terms involved
in the simulation of the mean soil moisture of the root zone, the cumulative evaporation, and the water
flux below the root zone upon leveraging on the SWAP model (van Dam et al., 2008). Their results
suggest uncertainty related to the crop parameters (i.e., crop height, root depth, and the Leaf Area
Index (*LAI*)) does not have a significant effect on these three model outputs in the setting analyzed.
Sobol indices estimated through a surrogate model are also used by Maina et al. (2022) to highlight
the significant impacts of hydrodynamic parameters' uncertainties on simulated evapotranspiration.
These Authors show that, under energy limited conditions and where plants have access to
groundwater, the uncertainty on evapotranspiration is related to uncertainties in saturated hydraulic
conductivities. Under water limited conditions, the parameters that contributes to the evaporation
uncertainty are those related to unsaturated flow conditions.
While the above-mentioned studies constitute only a sample across the broad literature associated with
diagnosis of LSMs through sensitivity analyses, they clearly show that the importance of model
parameters depends on several factors (such as, e.g., the target model output considered, the processes
embedded in the employed LSM, and the hydroclimatic conditions) and possibly on the selected
sensitivity analysis methodology.
This work aims at providing a comprehensive sensitivity analysis across spatial and temporal locations
within a hydrological system to highlight the most relevant model parameters and the corresponding
processes that need to be considered in a LSM. Here, we rely on a modular LSM developed at the
Institut Terre et Environnement de Strasbourg (ITES – Strasbourg Earth and Environment Institute) to
simulate key components of the water cycle (i.e., transpiration, evaporation, and groundwater
recharge) and to assess their sensitivity with respect to diverse model parameters that are typically



uncertain. We conduct a detailed sensitivity analysis by considering four diverse sensitivity indices:
(*i*) the distribution-based Borgonovo index (Borgonovo et al., 2007); (*ii*) the variance-based Sobol
indices (Sobol, 2001); and (*iii*) the moment-based *AMAE* and *AMAV* indices (Dell'Oca et al., 2017).
The joint use of these metrics is exemplified upon relying on realistic field conditions (in terms of,
e.g., climate, vegetation, and soil type) associated with two watersheds in the Vosges region (France)
across a one-year period. The relevance of relying on various sensitivity analysis, each providing a
unique contribution to enriching our knowledge of the system behavior, is underlined in several studies
(e.g., Maina and Guadagnini, 2018; Bianchi Janetti et al., 2019; Ju et al., 2021; Sandoval et al., 2022;
and references therein).
The methodological aspects associated with the LSM development and implementation, the definition
of the various sensitivity indices, and the description of the hydrological settings associated with the
catchments are presented in Section 2. Modeling results and the ensuing sensitivity analyses are
illustrated in Section 3, while conclusions are drawn in Section 4.
**2 Methodology**
**2.1 NIHM modular Land Surface Model**
The NIHM (Normally Integrated Hydrological Model) modular Land Surface model (NIHM-MLSM,
see Pan et al. (2015) and Jeannot et al. (2018)) is a numerical model design to compute on an hourly
basis (*i*) the energy balance at the soil and vegetation (vegetation being considered as a single layer)
surfaces, as well as, (*ii*) the water balance from the top of the vegetation layer to the groundwater table.
Diverse mathematical formulations for processes such as transpiration, evaporation and snow melt,
can be selected in conjunction with a modular structure on the basis of the observation that (*a*)
application of a unique model formulation across different soil and vegetation types is questionable
(Hogue et al., 2006) and (*b*) this allows adaptation to system complexity (Fisher and Koven, 2020).





Details of NIHM-MLSM are provided in the supplementary material. Here, we recall only the main
mathematical formulations, assumptions and parametrization.

*2.1.1 Energy balance*
The different components of the energy balance for the surface near the canopy layer are:
$$
\begin{cases}
R_n = R_{S\downarrow}(1-\alpha_c)\left(1-e^{-K_{ext}LAI}\right)+\varepsilon_c R_{L\downarrow}-\varepsilon_c\zeta T_c^4 \\[2mm]
H = \dfrac{\rho_a c_a}{r_{ac}}\left(T_c - T_a\right) \\[2mm]
\rho_w \lambda Tr = \dfrac{\rho_a c_a}{\gamma(r_{ac}+r_c)}\left[e_s^{sat}\left(T_c\right)-e_s\right]
\end{cases}
. \qquad (1)
$$


The corresponding components for the soil surface are:
$$
\begin{cases}
R_{n,s} = R_{S\downarrow}(1-\alpha_s)e^{-K_{ext}LAI}+\varepsilon_s R_{L\downarrow}-\varepsilon_s\zeta T_s^4 \\[2mm]
G = \dfrac{\rho_a c_a}{r_g}\left(T_s - T_g\right) \\[2mm]
H = \dfrac{\rho_a c_a}{r_{as}}\left(T_s - T_a\right) \\[2mm]
\rho_w \lambda E = \dfrac{\rho_a c_a}{\gamma r_{as}}\left(e_s^{sat}\left(T_s\right)-e_s\right)
\end{cases}
. \qquad (2)
$$


Here, $R_n$ [W.m$^{-2}$] is net radiation at the surface; $\rho_w \lambda E$ and $\rho_w \lambda Tr$ [W.m$^{-2}$] are surface latent heat flux
related to evaporation and transpiration, respectively; $H$ [W.m$^{-2}$] represents sensible heat flux (also
termed conductive heat flux) between the surface and the atmosphere; $G$ [W.m$^{-2}$] is the conductive
heat flux between the soil surface and the underground; $R_{S\downarrow}$ [W.m$^{-2}$] and $R_{L\downarrow}$ [W.m$^{-2}$] are the
incoming solar radiation and the longwave radiation, respectively; $\alpha_s$ is the soil albedo (-); $K_{ext}$ is the
canopy attenuation coefficient [-]; $LAI$ is the leaf area index [-]; $\varepsilon_s$ is soil emissivity [-]; $\zeta$ is the





Boltzman constant [W/m/K$^4$ ]; $T_a$, $T_g$, and $T_S$ are the air, underground and soil temperature [K],
respectively; $\rho_w$ and $\rho_a$ are water and air density [kg/m$^3$] respectively; $c_a$ is the specific heat of dry
air at constant pressure [J/kg/K]; $r_g$ and $r_{as}$ are the aerodynamic resistance at the soil and canopy
surface, respectively [s/m]; $e_s$ and $e_s^{sat}$ are the water vapor pressure and the water vapor pressure at
saturation [Pa], respectively; $\gamma$ is the psychrometric constant [Pa/K]; and $\lambda$ is the latent heat of water
vaporization [J/Kg].

Main assumptions related to the formulation of the energy balance comprise the following:

-    steady-state is considered, upon assuming that vegetation and soil layers have negligible heat

capacity;

-    conductive heat fluxes are expressed on the basis of a resistance analogy, similar to Ohm's law;

-    the amount of energy absorbed by the vegetation and received by the ground are estimated by

assuming a Beer-Lambert transmission reflectivity through the vegetation (Deardorff, 1978;

Taconet et al., 1986) and depend on the leaf area index (*LAI*) and an attenuation coefficient

(hereafter denoted as $K_{ext}$);

-    transpiration takes place only in the canopy; stomatal conductance is evaluated using a Jarvis-

type multiplicative model (Cox et al., 1998; Jarvis, 1976) and is affected by the environmental

factors embedded in the efficiency functions (solar radiation, air temperature, vapor pressure

deficit); the *LAI* is used to scale stomatal conductance to canopy conductance;

-    water intercepted by the canopy is assumed to evaporate with negligible impact on energy

balance (Kergoat, 1998);

-    the soil heat flux is approximated as proportional to the net radiation (Clothier et al., 1986;

Choudhury and Monteith, 1988; Kustas and Daughtry, 1990); for this study, the coefficient of

proportionality between the former and the latter is set at 0.5 $\gamma$ (Singh and Sharma, 2017,

Norman et al., 1995; Anderson et al., 1997; Boegh et al., 2000).



The equations governing energy mass balance are solved upon considering the surface temperature as
unknown and using a Newton Raphson method. If convergence is not reached after a maximum
number of user-defined iterations, temperature is set to corresponding value associated with the
previous time step. This approximation is assumed to be appropriate due to the small time steps
employed (hourly time steps).

### *2.1.2 Water flow and balance*
Water balance is formulated for three diverse compartments, i.e., the canopy, the snow cover, and the
soil. Key concepts associated with the water balance model for the canopy are: (*i*) water from
precipitation is partly stored in the canopy, whose storage capacity is limited to a maximum value; (*ii*)
the intercepted water is subject to evaporation and does not contribute to throughfall (i.e., the process
according to which excess water leaves wet leaves to reach the ground surface).
The snow model is adapted from the snow module of the HBV hydrological model (Seibert and
Bergström, 2022; Seibert and Vis, 2012). It consists in splitting precipitation (after interception) in
either snow, rain, or both. A conceptual model based on snowpack temperature is used to estimate
snowmelt fluxes (Neitsch et al., 2002).
Flow in the unsaturated zone is described by introducing three types of reservoirs (or layers): (*i*) the
litter, corresponding to the layer in contact with the atmosphere and where only evaporation takes
place; (*ii*) the root zone, which is colonized by plant roots and supplies water for transpiration; and (*iii*)
a set of sequential reservoirs to mimic vertical water movement below the root zone down to the
groundwater table. Each reservoir is defined through a given water content at saturation, the water
content at wilting point (which is also considered as the residual water content), and water content at
field capacity.
Water from throughfall and melted snow infiltrates in the litter layer. Evaporation (as computed by
energy balance at the soil surface) occurs only in this layer, and the amount of evaporated water is
linearly related to water content. Water drained from the litter layer enters the root layer. Transpiration





(estimated with the energy balance for the canopy) takes place only in this layer, and its amount is
adapted according to the available water therein. Drainage from the different layers is estimated in two
ways: (*i*) the water volume above the layer field capacity is drained immediately to the next layer, to
represent water movement due to gravity; and (*ii*) when water content lies between the field capacity
and the wilting point (i.e., residual water content), drainage is computed as an exponential function of
the available water amount.
Similar to other LSMs, NIHM-LSM requires the estimation of numerous forcing terms and parameters
related to climatic conditions, vegetation and soil characteristics. Several factors limit our ability to
obtain a reliable estimate of these forcing terms and parameters. These include, e.g., incompatibility
between the model scale and the support volume of the measurement and the inherent space and time
variability of most of the parameters that makes the exhaustive knowledge of model parameters and
forcing terms as practically unfeasible. Therefore, identification of the parameters that can be
considered as most *important* to given model outputs is critical to effectively assist modeling and
estimation of land surface energy and water fluxes. Note that we consider as *important* (or influential)
those model parameters whose variations impact to some extent model outputs of interest, i.e.,
transpiration, evaporation and groundwater recharge fluxes in this study.
Such parameters are identified through a global sensitivity analysis in an *ab initio* context, i.e., the
degree of uncertainty assigned to the vegetation and soil model parameters is grounded on a priori
qualitative knowledges (e.g., prior experience, literature data). In the present study, we do not evaluate
parameter uncertainty based on a model calibration procedure against experimental data associated
with the modeled system (e.g., measured transpiration fluxes).

## 2.2 Global Sensitivity analysis

A critical step in diagnosing and understanding the functioning of a model works involves quantifying
the relevance that different uncertain model parameters exert on the model results of interest to identify





the (possible) influential and non-influential parameter sets. These are here assessed through global
sensitivity analysis. In broad terms, the latter enables one to quantify the (relative) strength of the
influence of the variability/uncertainty in a given parameter on the corresponding
variability/uncertainty in the output(s) of the model analyzed.
Here, we rely on two complementary global sensitivity analysis methodologies: (*i*) density function-
and (*ii*) and moment-based strategies. While the former is tailored to analyze the effects that variations
of uncertain model parameters have on the whole (probability or cumulative) density function of the
model output, the latter focuses on the impact on given statistical moments of the density function of
model output. Here, we consider the Borgonovo index (Borgonovo et al., 2007) as a density function-
based metric. For moment-based metrics, we use the Sobol indices (Sobol, 2001) and the *AMAE* and
*AMAV* indices (Dell'Oca et al., 2017).
Considering *X* as a set of random independent parameters and *Y* as the corresponding model output,
the Borgonovo index ( *B* ) associated with parameter $X_i$ is defined as:

$$B_{X_i} = \frac{1}{2} \int f_{X_i}(x_i) \left[ \int \left| f_Y(y) - f_{Y|X_i}(y) \right| dy \right] dx_i \qquad (3)$$

Here, $f_{X_i}(x_i)$ is the marginal probability density function (pdf) of the *i*-th model (input) parameter $X_i$;
$f_Y(y)$ and $f_{Y|X_i}(y)$ are the unconditional and conditional (to a given value of $X_i$) marginal pdf of *Y*,
respectively. Note that the Borgonovo index grounds the concept of the sensitivity of *Y* to $X_i$ on the
base of the (average) distance between the unconditional pdf of the output and its counterparts
stemming from conditioning on diverse plausible values of $X_i$. This index ranges in the unit interval,
where a null value corresponds to scenario in which the pdf of *Y* is unaffected by variations in
parameter $X_i$.





We also rely on the classical Sobol indices (Sobol, 2001) to quantify the contribution of the uncertainty
in $X_i$ to the model output variance when considered alone, *i.e.*, principal index $SP_{X_i}$, or as it interacts
with other parameters, i.e., total index $ST_{X_i}$. The principal Sobol index associated with $X_i$ is given by:

$$SP_{X_i} = \frac{V\left[E\left[y \mid X_i\right]\right]}{V\left[y\right]}$$   (4)

where $E[\cdot]$ and $V[\cdot]$ denote the expectation and variance operators, respectively, and $E[y|X_i]$ is the
expected value of $Y$ conditional to a particular value of $X_i$. The principal Sobol index measures the
relative contribution of $X_i$ to the model output variance without considering interactions with other
uncertain model parameters. The corresponding total Sobol index embeds also the contributions of
interactions with the remaining model parameters and is defined as:

$$ST_{X_i} = SP_{X_i} + \sum_{x_j} SP_{X_i, X_j} + \sum_{x_j, x_k} SP_{X_i, X_j, X_k} + ...$$   (5)

where $SP_{X_i, X_j}$ is the fraction of model output variance due to the interactions between parameters $X_i$
and $X_j$ (the remaining symbols being characterized by a corresponding meaning). We recall that the
total Sobol index represents the expected contribution of $X_i$ to the variance of the model output,
including contributions caused by its interactions with other input variables. Sobol indices are broadly
used because of their simplicity and intuitive nature to assess sensitivity of models to input parameters
by decomposing the total variance of a model output of interest into different contributions, each
associated with a subset of parameters. These indices are used to measure the importance of individual
parameters and interactions between parameters (Sobol, 2001).




To complement our investigation, we evaluate the moment-based metric introduced by Dell'Oca et al.
(2017), termed *AMA* indices. The latter quantify sensitivity as the degree of variations in given
statistical moments of the target model output *Y* that are due to the variability in model parameter $X_i$.
Considering the expected value of *Y*, we introduce the following moment-based index:

$$AMAE_{X_i} = \begin{cases} E\left[\left|y_0 - E\left[y \mid X_i\right]\right|\right]/\left|y_0\right| & y_0 \neq 0 \\ E\left[\left|E\left[y \mid X_i\right]\right|\right] & y_0 = 0 \end{cases} \qquad (6)$$

where $y_0$ is the unconditional expected value of *Y*. Considering the second (centred) statistical moment,
i.e., the variance of *Y*, we introduce following index:

$$AMAV_{X_i} = E\left[\left|V[y] - V\left[y \mid X_i\right]\right|\right]/V[y] \qquad (7)$$

Relying on the *AMA* indices enables one to assess the sensitivity of *Y* in terms of various salient features
of the probability density function of the target model output, as rendered through diverse statistical
moments. Here, we focus on the mean and the variance of the model output. These metrics have been
applied in diverse settings, including scenarios related to, e.g., groundwater hydrology (Bianchi Janetti
et al., 2019; Dell'Oca, 2023), subsurface energy resources associated with gas flow migration across
low-permeability media (Sandoval et al., 2022), analysis of seismic metabarriers (Zeighami et al.,
2023), dynamics of emerging contaminants in groundwater (Ceresa et al., 2023), or assessment of
infiltration structures (Dell'Oca et al., 2023).
Our reliance on various sensitivity indices is in line with the observation that it is often difficult for
one method to provide a complete sensitivity assessment. This is even more critical for complex
hydrological systems of the kind we consider here (Mai et al., 2022).





## 2.3 Study catchments and data-sets


The NIHM-MLSM model is run under realistic field conditions (in terms of climate, vegetation, and
soil type) on two catchments in the Vosges region of northeastern France (i.e., in the Bruche and Doller
catchments) that are characterized by similar climatic conditions while being associated with differing
soil types and vegetation. While the model is run in a distributed way on the whole extent of each
catchment, results are only illustrated for a selected location (or computational pixel) for each
catchment, for simplicity. Selection of each of these pixels is based on the criterion that they are
considered as a representative of the conditions associated with the corresponding catchment in terms
of soil type, climate, and vegetation cover. Both locations are subject to an oceanic climate, being
affected by continental traits (Peel et al., 2007) due to the action of Foehn. Consequently, considerable
fluctuations in local climatic variables, such as air temperature or rainfall rates, are experienced.
Historical streamflow data indicate a low-water period taking place between June and October and a
high-water period between December and March (Banque HYDRO, 2020).
The first exemplary location considered in this study is located in the Bruche catchment, which is
characterized only by vineyards on Calcosol soil. The second location considered is representative of
the Doller catchment, which is covered by deciduous forest, moorland and heathland in combination
with the Alocrisols soil. Figure 1 and 2 depict records of the main climatic forcings monitored across
the study period, i.e., precipitation, temperature, wind speed, and solar radiation reaching the canopy
for the Bruche and Doller watersheds, respectively.

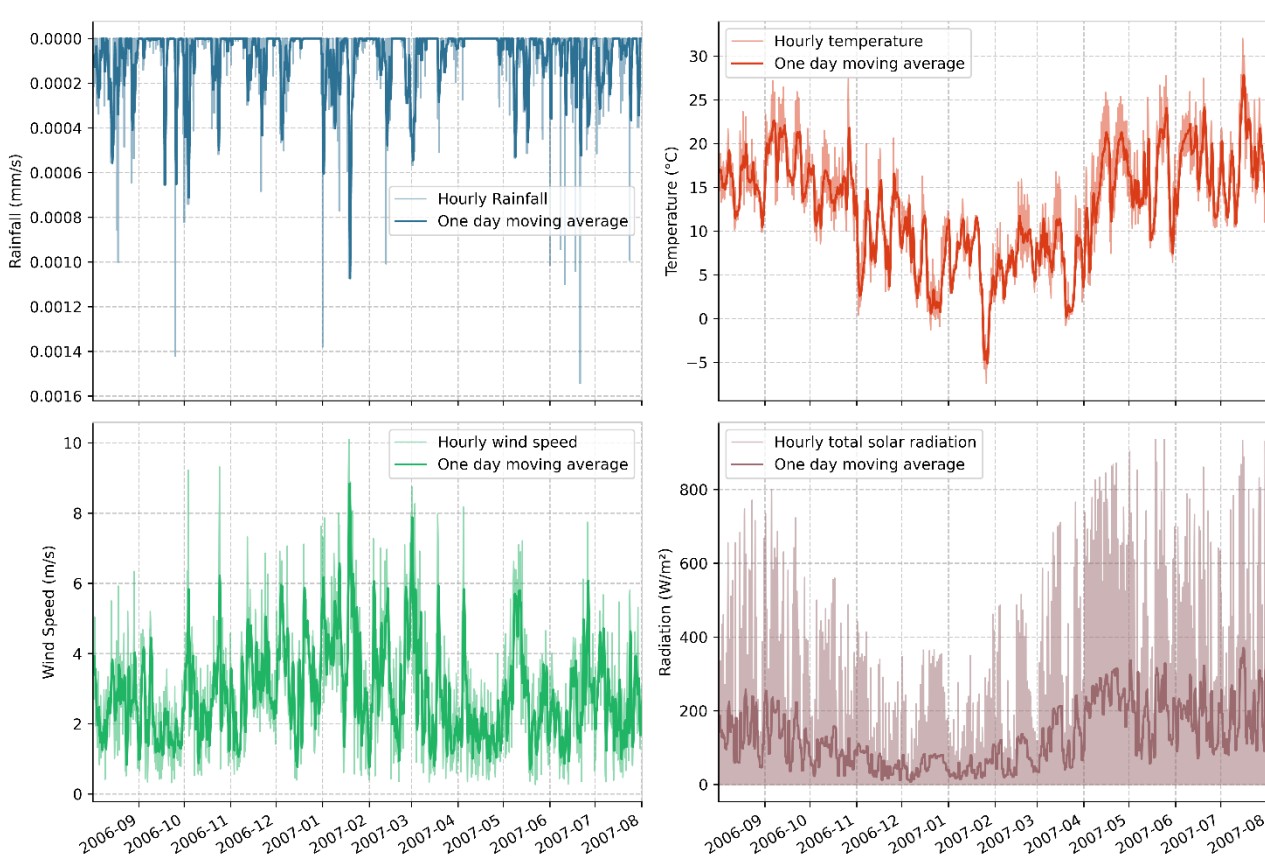

Figure 1: Main climatic forcing for the Bruche watershed.



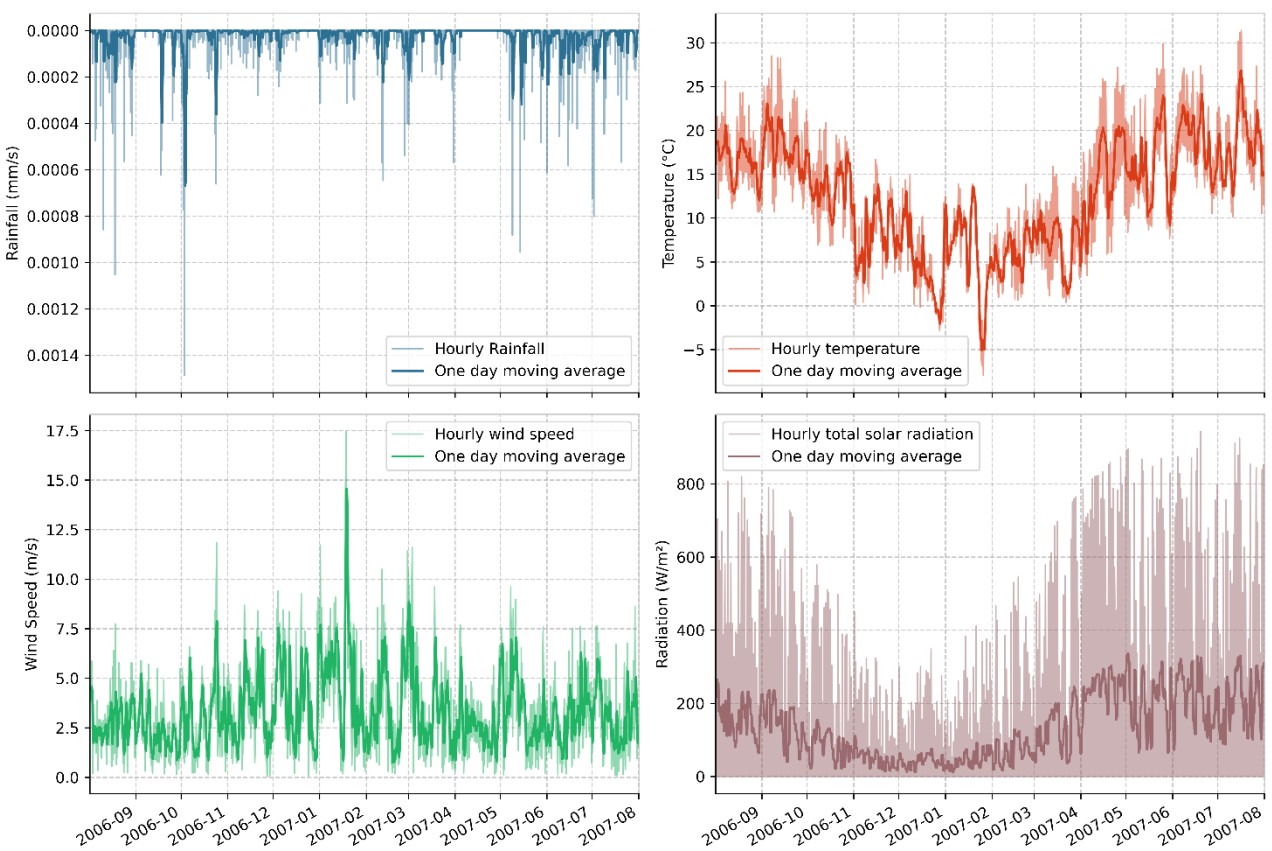


Figure 2: Main climatic forcing for the Doller watershed.


Climatic data (air temperature, air humidity, precipitation, snow, wind speed, incoming solar radiation, and longwave radiation) are included in the Safran database produced by Météo-France (Durand et al., 1993; Habets et al., 2008). The Safran system interpolates key climatic variables from ground measurements on a fixed grid of $8 \times 8$ km$^2$ with a hourly temporal resolution (Durand et al., 2009; Quintana-Seguí et al., 2008). It has been widely used to address hydrological monitoring and climate change studies (Vidal et al., 2010). Note that uncertainty on these forcing terms is not considered in this work which is otherwise specifically focused on the parameters required for LSMs.



In this study, the uncertain parameters involved in the evaluation of transpiration, evaporation, and

groundwater recharge fluxes at a given time are:

- the *LAI* and the albedo; considering their smooth variation over time, we rely on linearly
  interpolated monthly data; this implies that a given simulation time is associated with two *LAI*
  and two albedo parameter values;

- five vegetation-related parameters per vegetation type (i.e., precipitation interception, radiation
  attenuation, root depth, stomatal conductance, and canopy height);

- five parameters for the litter layer (i.e., residual water content, field capacity, porosity,
  thickness, and drainage coefficient) and four parameters for the root layer (residual water
  content, field capacity, porosity, and drainage coefficient, root depth being considered as a
  vegetation dependent parameter) per soil type.

A total number of 18 parameters is associated with a given type of vegetation and a given type of soil.

Monthly values for *LAI* and albedo are estimated from satellite data at a spatial scale of $3 \times 3$ km$^2$
(downloaded from https://land.copernicus.eu/global/products/ (Copernicus Climate Change Service,
2018)). When several values are associated with a given month, we consider their average as a
representative monthly value. If only one value is available, it is considered as the average value across
the month. Albedo values for the vegetation are computed upon relying on the albedo value rendered
by the satellite information and a prescribed albedo value for the soil assuming multireflection between
the soil and the vegetation (see Supplementary Material (SM)). Parameter uncertainty is also provided
in the dataset (in terms of a corresponding standard deviation).

The Corine Land Cover database (https://land.copernicus.eu/en/products/corine-land-cover) allows for
the identification of distinct vegetation categories at each studied catchment at raster scale of 100m
(European Union - SOeS, 2018). Table 1 lists the support (i.e., ranges of variability) associated with





the uncertain vegetation-related parameters for the diverse vegetation types related to the two
watersheds here considered. The width of these supports is identified on the basis of a detailed analysis
of previous literature studies. For completeness, we list the main literature sources analyzed for each
vegetation-related parameter:
- precipitation interception (Brecciaroli et al., 2012; Couturier and Ripley, 1973; Friesen and
Van Stan, 2019; Kergoat, 1998; Nicholas et al., 2011);

- radiation attenuation coefficient (Zhang et al., 2014);
- root depths (Escamilla et al., 1991; Freeling and Walbot, 1994; Leuschner et al., 2001; Mueller
et al., 2013; Richards, 2011; Grassland: Mission: Biomes, 2023);

- stomatal conductance (Gowdy et al., 2022; Brewer et al., 2022; Carter, 1998; Charreyron, 2011;
Hovenden and Brodribb, 2000; Jonard et al., 2011; Juan Carlos Baca Cabrera, 2021; Kim and
Verma, 1991; Mahhou et al., 2005; Mueller et al., 2013; Ocheltree et al., 2012; Reis and
Ribeiro, 2020; Song et al., 2018; Tardieu et al., 1991; Winkel and Rambal, 1993; Zhang et al.,
2012);

- canopy height (Campos et al., 2021; Liu et al., 2019; Matese et al., 2017; Grassland: Mission:
Biomes, 2023; Peiffer et al., 2014; Smirnova et al., 2008).








| Vegetation type | Interception [-] | | Attenuation [-] | | Root Depth [m] | | Stomatal conductance [m/s] | | Canopy height [m] | |
|---|---|---|---|---|---|---|---|---|---|---|
| **T1** | 0.1 | 0.4 | 0.16 | 0.54 | 0.5 | 2.5 | 0.005 | 0.015 | 0.2 | 1.2 |





| | | | | | | | | | | |
|---|---|---|---|---|---|---|---|---|---|---|
| **T2** | 0.1 | 0.53 | 0.29 | 0.65 | 0.6 | 2.3 | 0.0002 | 0.0036 | 12.6 | 27.0 |
| **T3** | 0.14 | 0.22 | 0.35 | 0.65 | 0.2 | 1.0 | 0.0011 | 0.0110 | 0.2 | 2.1 |

Table 1: Vegetation-dependent parameters (minimum and maximum values) for **T1:** Vineyards (Bruche catchment), **T2:** Deciduous forests (Doller catchment) and **T3:** Grasslands, Natural grasslands and pastures, Moors and heathland (Doller catchment).

Only the vineyards vegetation is considered (T1 in Table 1) at the Bruche catchment. Two types of vegetation are considered for the exemplary location selected at the Doller catchment. These correspond to (*i*) vegetation composed mainly of broad-leaved species, including shrub and bush understoreys for 2/3 of the pixel area (T2 in Table 1) and (*ii*) vegetation resulting mainly from forest degradation (low and closed cover, dominated by bushes, shrubs and herbaceous plants) for 1/3 of the pixel area (T3 in Table 1).

Soil types are classified upon relying on the Regional Soil Reference System for Alsace and Vosges (https://data.europa.eu/data/datasets/fr-341142131-araa_bdsol-alsace_250000_2011?locale=fr). Six main categories are identified (Chambre Régionale d'Agriculture Grand Est, 2011, 2015) and denoted according to the World Reference Base for Soil Resources (IUSS Working Group WRB, 2022).In this work we consider only the first two soil layers (i.e., litter and root zone) and groundwater recharge is assumed to coincide with drainage from the root zone, the overall thickness of the unsaturated zone being limited to a few meters. Table 2 lists ranges of variability associated with the uncertain soil-related parameters for the diverse soil types of interest. As stated above, the width of these supports is set on the basis of previous studies (Belfort et al., 2018; Clapp and Hornberger, 1978; Dingman, 2002). Note that soil types S1 and S2 constitute typical traits of the Bruche and Doller catchment, respectively.

| Soil Type | Root layer | | Litter layer | | |
|---|---|---|---|---|---|
| | Field capacity $\theta_c$ | Porosity $\theta_s$ | Field capacity $\theta_c$ | Porosity $\theta_s$ | Thickness (m) |





| | | | | | | | | | | |
|---|---|---|---|---|---|---|---|---|---|---|
| **S1** | 0.22 | 0.33 | 0.42 | 0.48 | 0.25 | 0.35 | 0.50 | 0.80 | 0.05 | 0.15 |
| **S2** | 0.12 | 0.16 | 0.40 | 0.45 | 0.17 | 0.25 | 0.50 | 0.80 | 0.05 | 0.15 |

Table 2: Soil-dependent parameters (minimum and maximum values) for **S1:** Calcosols and Calcisols
(Bruche catchment) and **S2:** Alocrisols (Doller catchment).

Drainage coefficients for both layers and all soil types are set to range between $1.0 \times 10^{-7}$ and $9.0 \times$
$10^{-7}$. This range of variability has been defined on the basis of the temporal pattern of groundwater
recharge fluxes obtained through preliminary model runs (details not shown). Residual water content
is fixed at 0.01 for all soil types.

Evaluation of the global sensitivity indices listed in Section 2.2 is performed through a numerical
Monte Carlo (MC) approach. Parameter values are randomly sampled by considering model
parameters as independent and identically distributed random variables, each characterized through a
uniform distribution with support given in Table 1 and 2. With reference to *LAI* and albedo, the semi-
width of the support is set to the value of the standard deviation provided in the Copernicus data sets.
Sobol indices are calculated upon considering the algorithm described in Saltelli (2007). A total of
147,500 and 168,000 simulations are performed for the Bruche and the Doller catchment, respectively.
The number of simulations is higher for the Doller catchment due to the additional vegetation type
(see Table 1).
The temporal window associated with our simulations spans a period of two years (01/09/2005 to
31/08/2007). The analyses target solely the second year of simulations, to minimize impacts of initial
conditions on model outputs.





**3 Results**
**3.1 Catchments behaviors**
Here, we illustrate the type of results obtained with the modeling study to assist grasping the overall
behavior of the systems and to provide a first quantitative appraisal of the nature of the available
observations and modeling outputs related to the complex hydrological systems analyzed. We rely on
graphical depictions rendered in terms of the expected value +/- one standard deviation of daily
averaged values grounded on the set of MC simulations for the period from August 1st, 2006 to July
31th, 2007. Temporal dynamics of actual evaporation, actual transpiration, and groundwater recharge
fluxes are provided in Figures 3 and 4 for the selected locations in the two watersheds (see Section
2.3), together with the corresponding observed rainfall series. Detailed quantitative results concerning
the main components of the water cycle are listed in Table 3 and 4 for the Bruche and Doller catchment,
respectively.

|  | **Autumn** | **Winter** | **Spring** | **Summer** | **Total** |
|---|---|---|---|---|---|
| Precipitation (mm) | 212.7 | 165.1 | 237.7 | 287.6 | 903.2 |
| *(%)* | *24* | *18* | *26* | *32* | *100* |
| Evaporation (mm) | 39.21 / 4.41 | 54.93 / 6.30 | 65.50 / 12.73 | 60.70 / 15.52 | 220.34 / 33.98 |
| *(%)* | *18* | *25* | *30* | *28* | *100* |
| Transpiration (mm) | 10.74 / 4.21 | 2.05 / 1.59 | 55.85 / 18.69 | 60.25 / 18.69 | 128.89 / 40.84 |
| *(%)* | *8* | *2* | *43* | *47* | *100* |
| Groundwater Recharge (mm) | 116.65 / 25.31 | 75.00 / 7.98 | 63.12 / 11.95 | 74.86 / 24.03 | 329.62 / 60.88 |
| *(%)* | *35* | *23* | *19* | *23* | *100* |

Table 3. Amount of water volume (in mm) for the different seasons and over the year for the Bruche
catchment. Values of transpiration, evaporation, and groundwater recharge are evaluated through the
NIHM-MLSM model (mean / standard deviation). Percentage values (%) are defined as the ratio
between seasonal values and their yearly counterparts.







|  | Autumn | Winter | Spring | Summer | Total |
|---|---|---|---|---|---|
| Precipitation (mm) | 566.4 | 780.03 | 414.54 | 780.63 | 2541.3 |
| (%) | *22* | *31* | *16* | *32* | *100* |
| Evaporation (mm) | 34.45 / 1.52 | 71.22 / 3.63 | 43.17 / 8.47 | 25.69 / 2.53 | 174.54 / 13.71 |
| (%) | *20* | *41* | *25* | *15* | *100* |
| Transpiration (mm) | 15.4 / 4.06 | 1.60 / 0.71 | 44.22 / 12.63 | 58.64 / 14.2 | 119.86 / 30.21 |
| (%) | *13* | *1* | *37* | *49* | *100* |
| Groundwater Recharge (mm) | 315.19 / 80.72 | 609.40 / 24.83 | 201.77 / 35.9 | 246.20 / 105.8 | 1372.56 / 236.23 |
| (%) | *23* | *44* | *15* | *18* | *100* |

Table 4. Amount of water volume (in mm) for the different seasons and over the year for the Doller catchment. Values of transpiration, evaporation, and groundwater recharge are evaluated through the NIHM-MLSM model (mean / standard deviation). Percentage values (%) are defined as the ratio between seasonal values and their yearly counterparts.




Figure 3 Observed (a) precipitation, (b) calculated evaporation and (c) transpiration together with (c)

groundwater recharge at the Bruche watershed.




Figure 4. Observed (a) precipitation, (b) calculated evaporation and (c) transpiration together with (c)

groundwater recharge at the Doller watershed (note the different scale for recharge





Despite the geographical proximity of the watersheds, precipitation patterns are quite different. Tables
3-4 indicate that the annual amount of precipitated water over the year is very different between the
two catchments (903.2 mm for the Bruche and 2541.3 mm for the Doller) and that during Winter the
Bruche catchment is relatively dry, while the Doller catchment experiences significant precipitation
events. These findings are further corroborated by the inspection of Fig.s 3a, 4a.

Tables 3-4 suggest that evaporation in both catchments occurs over the whole year, with very similar
total amount of evaporated water. However, inspection of Figs. 3b, 4b and of the seasonal values listed
in Table 3-4, reveals that evaporation intensify during Winter for the Doller catchment, due to more
frequent and intense precipitation (in line with the previous observation regarding the difference in the
precipitation patterns during Winter). Additionally, despite the stronger similarity in the precipitation
patterns during Summer (see also percentage values in Table 3-4) evaporation at the Doller catchment
appears to be less pronounced than at Bruche. We attribute this difference to the (overall) lower values
of the attenuation coefficient (characteristic of vineyard vegetation cover) at Bruche with respect to
the counterparts at Doller (see Table 1). The periods with the highest uncertainties in the evaporation
(as quantified in terms of standard deviation of model outputs) are generally observed to take place
between rainy episodes (at both catchments) and during Summer at Bruche, when a significant amount
of solar radiation is intercepted by the vineyard that is characterize by a more uncertain attenuation
coefficient that the vegetation covers present at Doller (see Table 1).
Joint inspection of Tables 3-4 and Figs. 3c and 4c highlights that transpiration fluxes at both
catchments are characterized by a typical seasonal variability, with very poor transpiration fluxes in
Winter (less than 2% of the annual transpired water) and increased transpiration (close to 50%) in
Summer. These findings are in line with the weather conditions (temperature, radiation) and vegetation
status. When active, transpiration is more intense in the Bruche watershed due to its higher soil storage
capacity that allows for water extraction in the root zone to fulfil the evaporation potential. Notably,





uncertainty in the transpiration is larger in correspondence of the growing phase of vegetation during
Spring at both watersheds.

Comparison of Tables 3-4 reveals different (relative) amounts of groundwater recharge at the two
watersheds: groundwater recharge represents about 36% and 54% of the yearly precipitation for the
Bruche and Doller catchment, respectively. We mainly ascribe these differences to the diverse values
of the soil field capacity (see Table 2) at the two catchments (recall that the amount of water above
field capacity constitutes the groundwater recharge at a given time step). Such element also implies
very different patterns (see Fig.s 3d, 4d) in the behavior of the groundwater recharge at the two
catchments: the higher values of the Bruche soil field capacity result in smoother temporal fluctuations
of the groundwater recharge, while the lower values of the field capacity for the Doller catchment yield
a higher reactivity. At the same time, groundwater recharge mostly occurs during autumn and Winter
at both catchments while still remaining significant also during Summer (around 20% of the annual
recharge). Notably, in both catchments the uncertainty in the groundwater recharge tends to increase
with the expected value of the latter.

The results from the Monte Carlo simulations can also be analyzed in terms of the ensuing pdf of an
output of interest at a given time. Figure 5 depicts exemplary pdfs obtained for diverse model outputs,
i.e., (a) evaporation, (b) transpiration, (c) groundwater recharge at Bruche, (d) groundwater recharge
at Doller, at different times, considering unconditional results (red curves) and conditioning on diverse
subintervals of variability for a given parameter. With reference to the latter element, we select here
five equiprobable subintervals, for (a) litter drainage, (b) albedo coefficient, (c) root drainage rate and
(d) root depth. This type of visual analysis is akin to a regionalized sensitivity analysis. It helps one to
grasp the impact that conditioning on diverse values (comprised within subintervals according to which
the overall support is partitioned) of a parameter might have on the pdf of an output of interest.




Figure 5. Probability density functions related to: (a) evaporation on July 7[th], 4 p.m. at Bruche with
prescribed litter drainage rate; (b) transpiration on December 31[th], 12 a.m. at Bruche with prescribed
albedo coefficients; (c) groundwater recharge on October 5[th], 12 a.m. at Bruche with prescribed root
drainage rate(d) groundwater recharge on February, 2[nd], 12 a.m. at Doller with prescribed root depth.
In each panel, we consider the unconditional (red curve) pdf of each output and its counterparts





conditioned (blue curves) on five different (equally probable) subintervals (middle bin conditioning
(denoted as Cond.) value provided in the legend) according to which the support of a given parameter
is partitioned.

Inspection of Figure 5 reveals several interesting features. The pdfs of the actual evaporation rate, as
recorded on September 7[th] at 4.00 p.m. at Bruche, visually resemble a Gaussian distribution (with a
slight asymmetry) and conditioning on smaller litter drainage values results in lower average
evaporation rates and higher variance (see Fig. 5a). Null values of the actual transpiration rate are
generally likely to occur during the December 31[th] at 12 a.m. at Bruche, while greater values of the
albedo coefficient lead to higher average and variance (see Fig. 5b). Inspection of the pdfs of the
groundwater recharge, as recorded on May 10[th] at 12.00 a.m. at Bruche (Fig. 5c), suggests a strong
sensitivity to the root drainage coefficient. On the other hand, the pdf of the groundwater recharge, as
recorded on February 11[th] at 12.00 a.m. at Doller, exhibits a bimodal behavior and appears to be
insensitive to the root depth (Fig. 5d).

These preliminary investigations for the diverse outputs and their response to model parameter
variations suggest a complex behavior of the LSM here investigated. A quantitative appraisal of
sensitivity is illustrated in Section 3.2 on the basis of the metrics introduced in Section 2.2.

**3.2 Global Sensitivity Analysis.**
We compute the global sensitivity indices introduced in Section 2.2 on an hourly basis over a temporal
window of one year. Figure 6 depicts color-coded (from red/high to white/low) values of $B$ (Eq. 3) $ST$
(Eq. 5), $AMAE$ (Eq. 6), and $AMAV$ (Eq. 7) indices for the evaporation rate at Bruche across the year
and the diverse model parameters (listed along the vertical axis; corresponding parameter identification
number is defined in the Symbol List of the Supplementary Material). Figure 7 depicts corresponding





results for the Doller catchment. Fig.s 8-11 are patterned after Fig.s 6-7 considering the results for
transpiration rate and groundwater recharge.

We recall that, *B* (similar to all other indices) should vanish for certain outputs of interest that one
knows are for sure insensitive to certain parameters. For example, the value of *B* for the evaporation
at Bruche during the month of July associated with the *LAI* of January must be zero. However,
inspection of Fig. 6a does not reflect this anticipated outcome. This apparent anomaly is attributed to
a random noise (that would require a markedly high amount of additional computation hours to be
reduced) stemming from the still incomplete sampling of the parameters space (despite our analysis
incorporates an extensive number of random samples). Drawing from these findings, we identify a
threshold value of $B = 0.17$ (average value in correspondence of instances associated with an expected
null value of *B* ) as a benchmark for evaluating the adequacy of parameter sampling within our
sensitivity analysis. Consequently, we disregard from our sensitivity analysis instances in which *B*
falls below 0.17, i.e., we assign a value of zero to *B*, *ST*, *AMAV*, and *AMAE* to enhance interpretability
of visual representations.
Overall, the evaporation rate in the two catchments (see Fig.s 6 and 7) is mainly sensitive to the
characteristics of the litter layer (in terms of layer thickness, field capacity, and drainage rate), the
amount of radiation reaching the soil surface (as expressed through the attenuation coefficient), and
the *LAI*. The relative influence of parameters related to the litter layer is generally higher for the
variance (see *ST*, *AMAV*) than for the expected value (see *AMAE*) of evaporation in both catchments.
Additionally, if a parameter influences the expected value of the evaporation rate at a given period
during the year, it also influences its variance (the opposite not being generally observed).



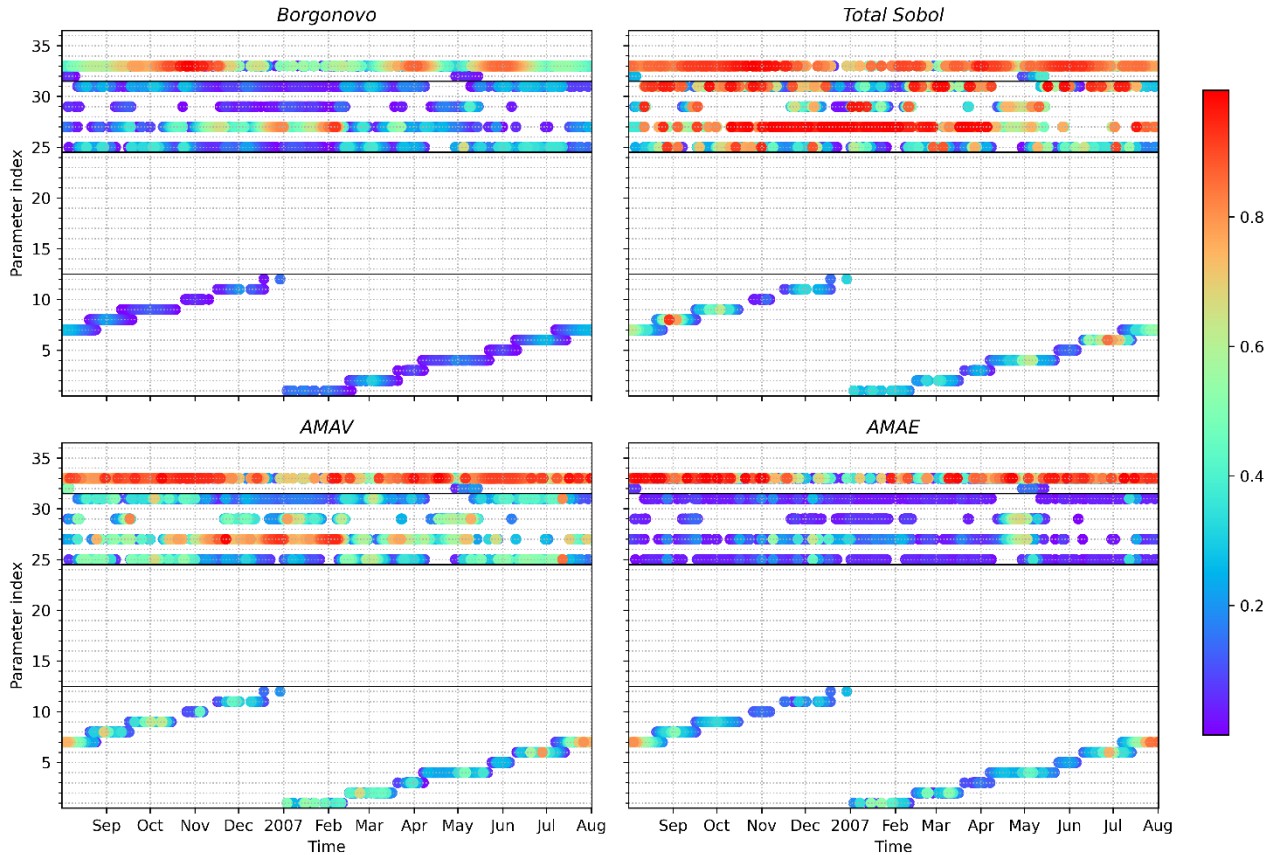


Figure 6. Temporal behavior of the sensitivity indices related to the evaporation rate at the Bruche
catchment. Parameter id from 1 to 12 corresponds to *LAI*; parameter id = 25 denotes root layer field
capacity, 27 is litter layer drainage coefficient, 29 is litter layer thickness, 31 is litter layer field
capacity, and 33 is attenuation coefficient (see Supplementary Material for the complete list of
parameter identifiers

565

Results for the indices associated with measurements of uncertainty of the output (i.e., *B* , *ST*, and

*AMAV*) at the Bruche catchment suggest the uncertainty in the evaporation rates switches from being

dominated by the litter layer drainage coefficient during Winter to being majorly influenced by the

variability in attenuation coefficient of the vegetation during the rest of the year. Considering the

sensitivity of the expected value of the evaporation rate (as rendered through *AMAE*), the attenuation





coefficient of the vegetation is the predominant parameter across the year with the exception of the
Winter season when the litter layer drainage coefficient gains relevance during a dry period in
December (see Fig. 1a), while the *LAI* becomes influential during the subsequent more wet period in
January. A similar pattern is documented also in correspondence of the dry month of April (here, also
the litter layer thickness gains some importance), which is then followed by the wet month of May.
Additionally, the *LAI* attains its highest influence (considering all of the sensitivity indices) during
Summer (July-September).



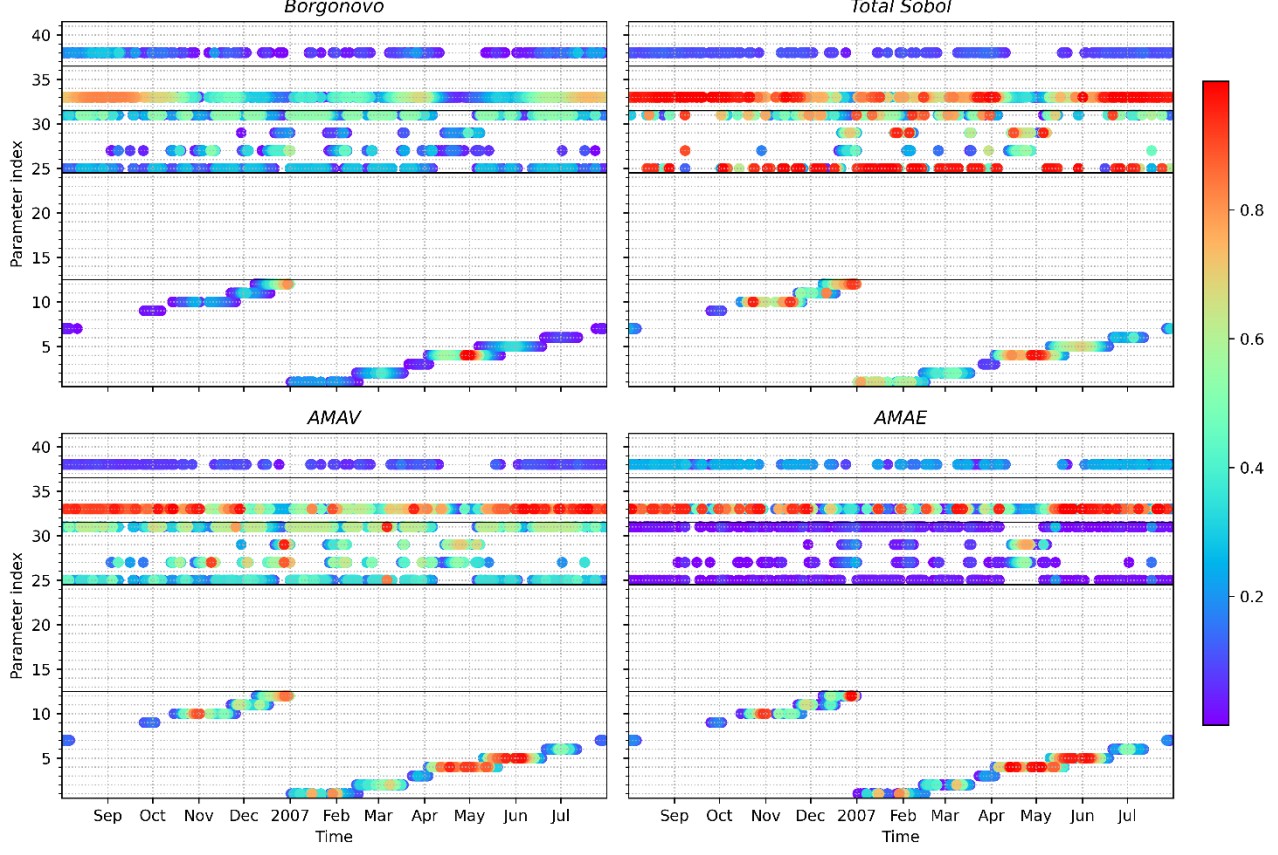






Figure 7. Temporal behavior of the sensitivity indices related to evaporation at the Doller catchment.
Parameter id from 1 to 12 correspond to *LAI*; parameter id = 25 corresponds to root layer field capacity,
27 is litter layer drainage coefficient, 29 is litter layer thickness, 31 is litter layer field capacity;
parameter id = 33 and 38 correspond to the attenuation coefficient for the two types of vegetation (see
Supplementary Material for the complete list of parameter identifiers

With reference to the Doller catchment, the attenuation coefficient of the deciduous forest (T2 in Table
2) has a strong influence over the uncertainty (i.e., $B$, $ST$, and $AMAV$) and the expected value (i.e.,
*AMAE*) of the transpiration rate, the dry periods during December and April being an exception. Note
that values of the corresponding indices for the other vegetation type (T3 in Table 2) shows a reduced
influence because it corresponds to only 1/3 of the land cover. At the same time, the soil layer
parameters that are most consistently influential to uncertainty of the evaporation rate across the year
are the litter and root layer field capacities, while they appear not to influence the expected value of
the evaporation rate at Doller. The drainage rate and thickness of the litter gain relevance only during
the no-rain period in April, jointly with the *LAI*, whereas the attenuation coefficient of deciduous forest
(T2 in Table 2) displays a reduced relevance. In contrast to Bruche, *LAI* is here mostly relevant during
October to June while being less relevant during Summer. Our results suggest that field capacity of
the litter and root layers are more relevant in the Doller than their counterparts in the Bruche watershed.
At the same time, the litter drainage coefficient is overall less relevant in Doller than in Bruche. Similar
to what we observe at Bruche, the litter field capacity is not influential when the rainy period starts
(i.e., in May).

With reference to transpiration (Fig.s 8 and 9), our results show that parameters related to vegetation
are characterized by a relative importance of that is significantly higher than their counterparts related
to soil. The relative influence of some parameters is sometimes higher when focusing on the expected
value of the transpiration rate than it is for its variance (compare *AMAE* to corresponding values of *ST*



and *AMAV*). This is clearly the case with the parameters that regulate the amount of energy reaching
the canopy, especially in Summer for *LAI* and in Winter for albedo.

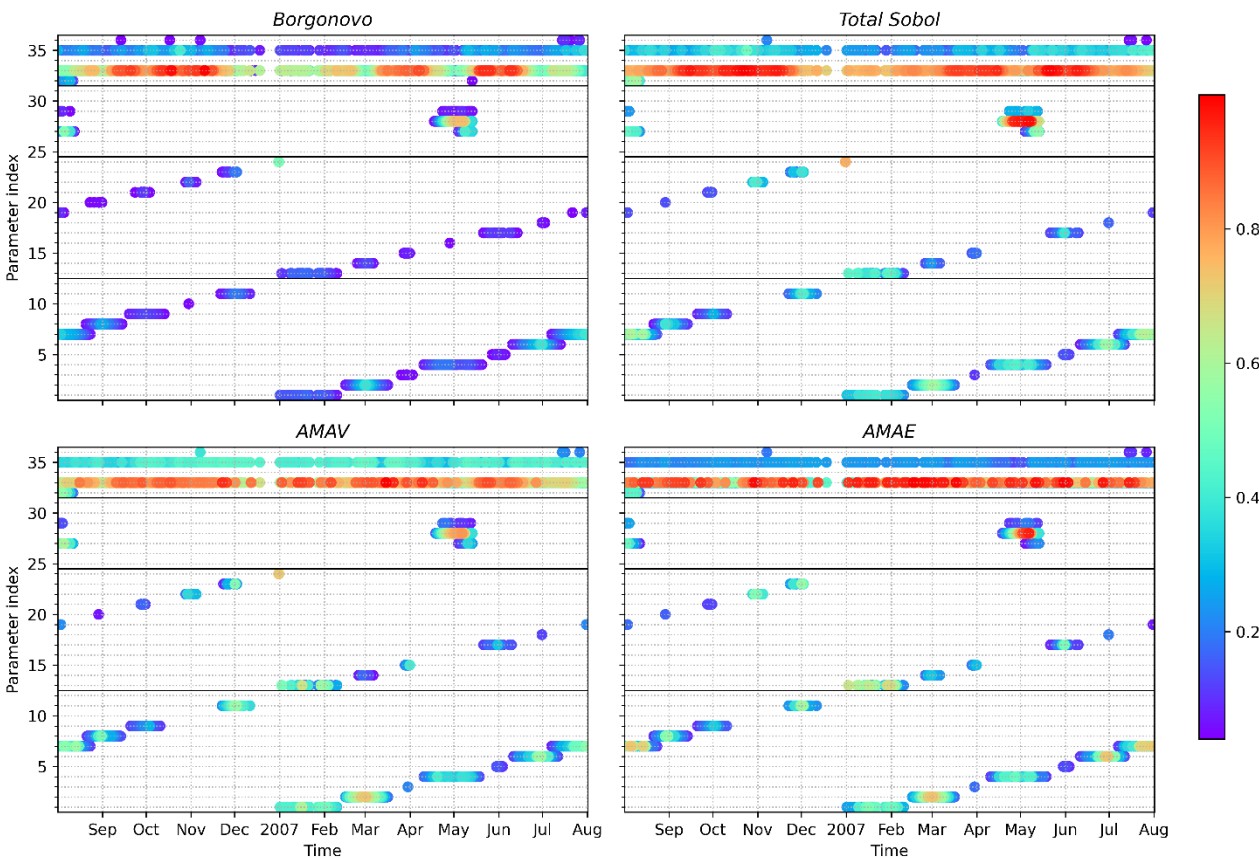


Figure 8. Temporal behavior of the sensitivity indices related to transpiration at the Bruche catchment.
Parameter id from 1 to 12 correspond to *LAI*; parameter id from 13 to 24 correspond to albedo;
parameter id = 33 and 35 correspond to the attenuation coefficient and to maximum stomatal
conductance, respectively (see Supplementary Material for the complete list of parameter identifiers).

Inspection of Fig. 8 highlights that transpiration in the Bruche catchment is overall sensitive to
vegetation-related parameters (chiefly to the attenuation coefficient and to stomatal conductance)
during the year. Otherwise, the litter layer drainage coefficient exhibits a strong influence during the





wet period in May that follows the no-rain period of April. The displayed sensitivity is consistent with
the observation that water available in the root layer for transpiration is supplied by the litter layer, by
drainage from the litter layer. At the same time, transpiration in Bruche exhibits a pattern of sensitivity
to all of the parameters associated with *LAI* during the year in close similarity to what we observe for
evaporation (see Fig. 6). Sensitivity to the albedo coefficient is mostly relevant during the Winter
period where solar radiation is quite limited. Interestingly, the impact of the parameters related to the
vegetation (attenuation coefficient and maximum stomatal conductance) on the expected value and
variance of the transpiration rate are different. The attenuation coefficient mainly affects the expected
value of transpiration (see corresponding values of *AMAE* in Fig. 8) as compared to its variance (see
corresponding values of *TS* or *AMAV* in Fig. 8), whereas the relative importance of stomatal
conductance is more marked for the variance than for the expected value.


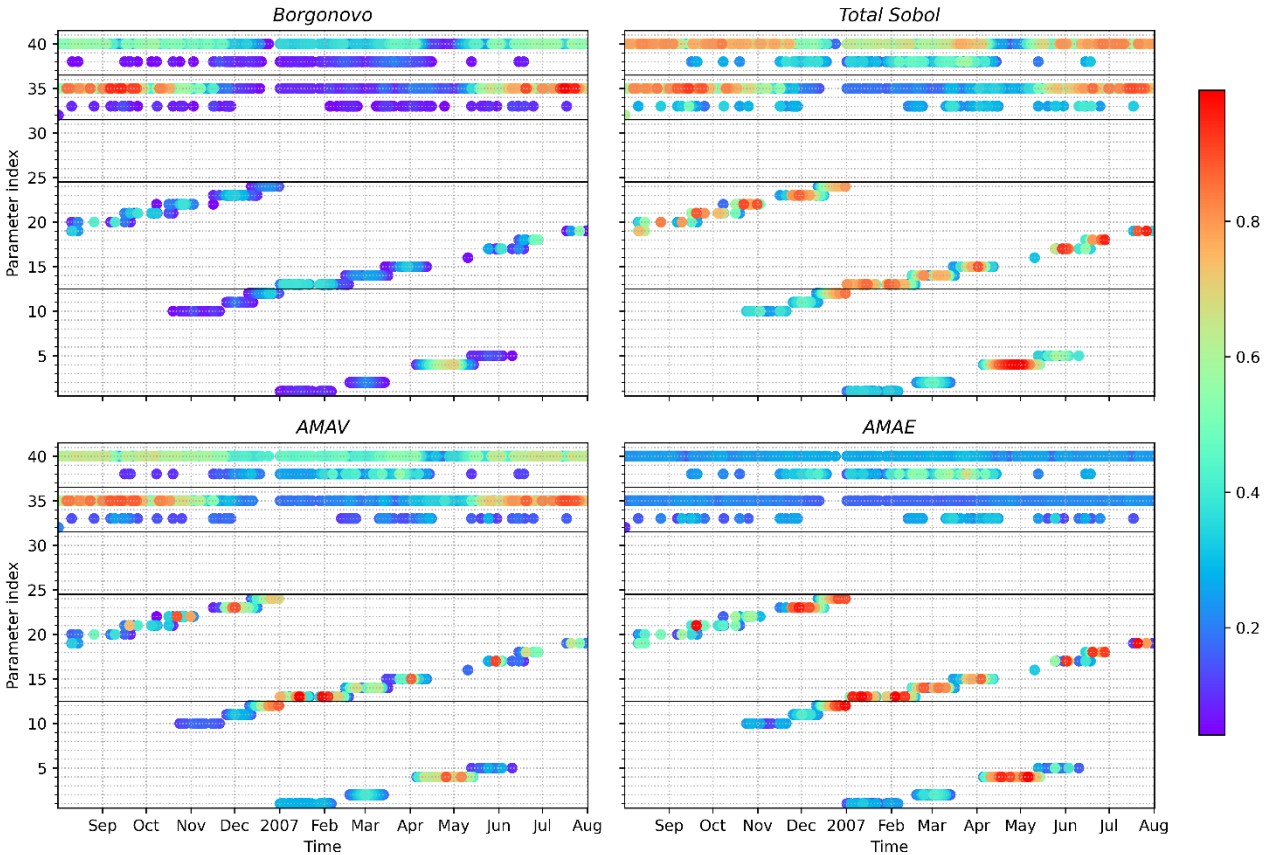

Figure 9. Temporal behavior of the sensitivity indices related to transpiration at the Doller catchment. Parameter id from 1 to 12 correspond to *LAI*; parameter id from 13 to 24 correspond to albedo; parameter id = 33 and 38 correspond to the attenuation coefficient of both vegetation types; parameter id = 35 and 40 correspond to maximum stomatal conductance of both vegetation types (see Supplementary Material for the complete list of parameter identifiers

Considering transpiration in the Doller catchment, Fig. 9 reveals that: (*i*) the *LAI* exerts a marked influence during the no-rain period of April and December (similar to the sensitivity of evaporation in Doller; see Fig. 7), while it is not influential during Summer; (*ii*) the albedo coefficient consistently impacts transpiration during the Winter-middle of Spring period (i.e., during low radiation periods) while the strength of its influence is more intermittent during the rest of the year; (*iii*) parameters





related to the soil layer have no influence on transpiration in Doller during the whole year; (*iv*) during
Winter, transpiration appears to be chiefly controlled by the albedo coefficient, the maximum stomatal
coefficient and the attenuation coefficient of the deciduous forest; (*v*) in contrast to Bruche, the
maximum stomatal conductance of the deciduous forests (T3 in Table 2) and of the degraded forest
(T2 in Table 2), are the most relevant vegetation-related parameters in Doller (the former being
especially relevant during Summer and fall, while the latter is more uniformly influential during the
year). The variation in the sensitivity of the transpiration rate to vegetation-related parameters across
the two catchments aligns with the distinct ranges of variability assigned to the stomatal conductance
among different types of vegetation. Specifically, the stomatal conductance of the forest (that
dominates at Doller) is relatively low as compared to that of vineyards (vegetation type of Bruche).

The results encapsulated in Figs. 10 and 11 surprisingly show that groundwater recharge is sensitive
to very few parameters. None of the vegetation related parameters are ranked as important on the basis
of the diverse sensitivity metrics here considered, intercept being the sole exception. Inspection of Fig.
10 highlights that groundwater recharge in the Bruche watershed can be considered as chiefly sensitive
to the root drainage coefficient and, albeit to a reduced extent, to the rainfall interception. In particular,
the root drainage coefficient is affecting the uncertainty in the groundwater recharge in Bruche in a
consistent manner when considering $B$, $ST$ and $AMAV$ (Fig. 10a-c). The same finding holds for the
sensitivity of groundwater recharge to the rainfall intercept (see Fig. 10a-c; note the zero values during
the no-rain period, as expected). Considering the expected value of groundwater recharge (see Fig.
10d), the root drainage coefficient shows a strong influence during the rain events of May (that follows
the no-rain period of April), while rainfall interception is generally less influent on the expected value
of the groundwater recharge during the whole year.





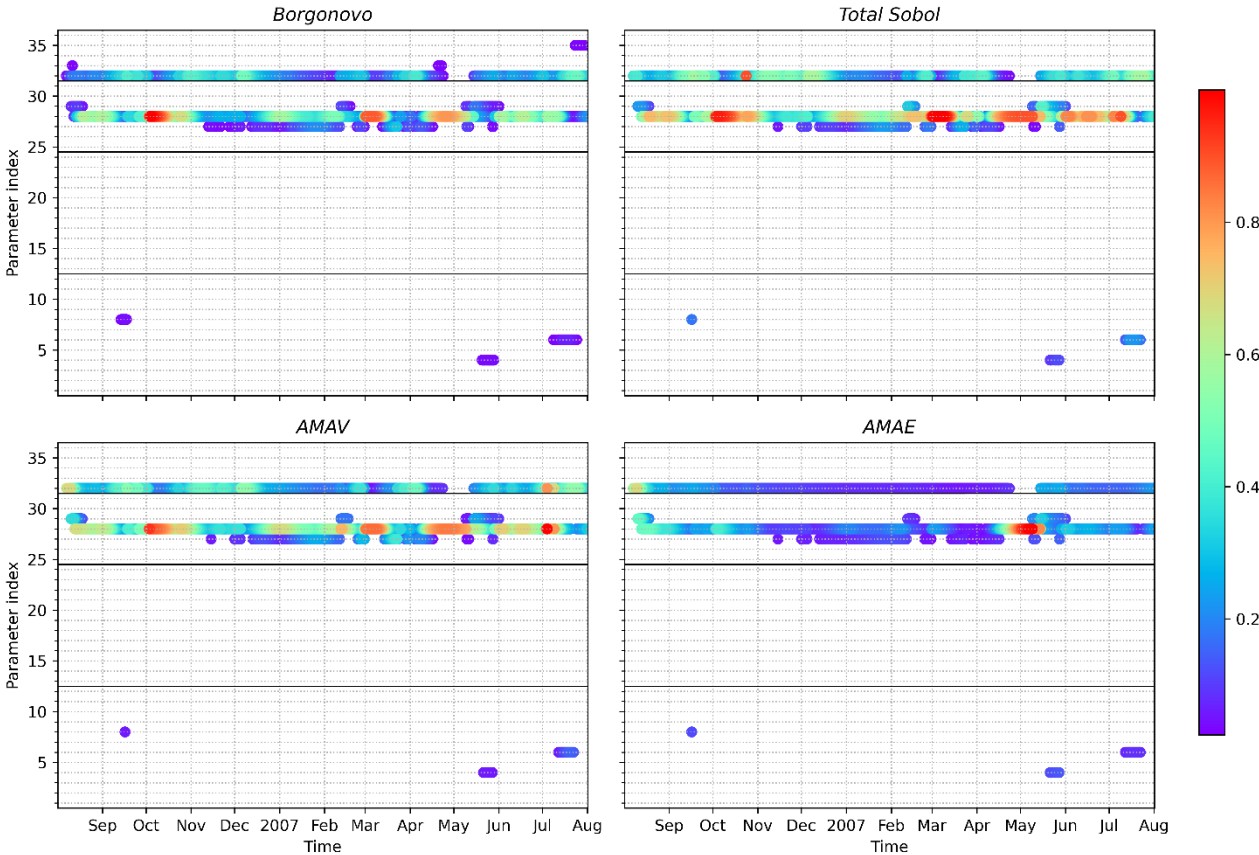

Figure 10. Temporal behavior of the sensitivity indices related to groundwater recharge at the Bruche

catchment. Parameter id 28 and 32 correspond to root drainage coefficient and rainfall interception,

respectively (see Supplementary Material for the complete list of parameter identifiers

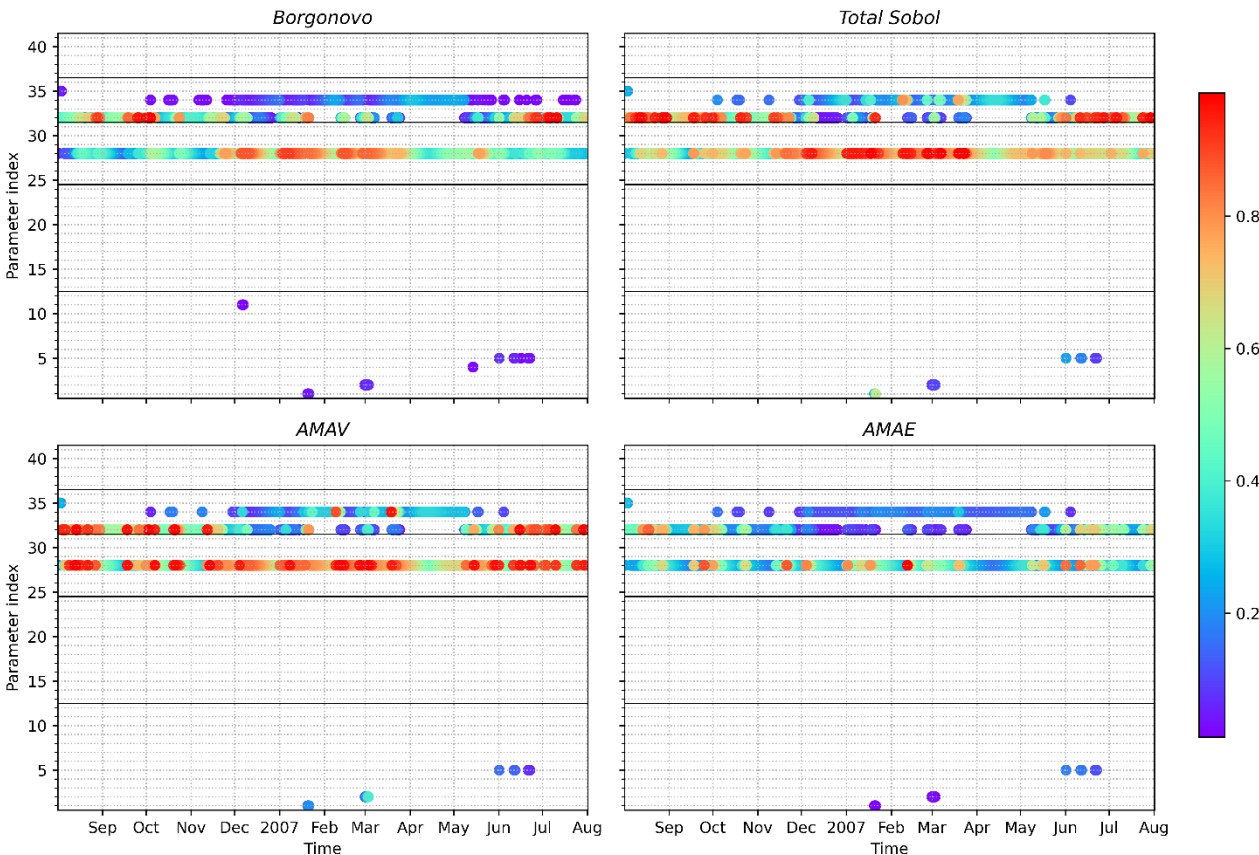

Figure 11. Temporal behavior of the sensitivity indices related to groundwater recharge at the Doller catchment. Parameter id 28, 32, and 34 correspond to root drainage coefficient, rainfall interception and root layer thickness, respectively (see Supplementary Material for the complete list of parameter identifiers.

Analysis of Fig. 11 highlights that groundwater recharge in the Doller is majorly sensitive to the root drainage coefficient and the rainfall interception. Rainfall interception by the canopy (T2 in Table 2) is dominant during Summer while the root zone drainage rate plays an enhanced role in Winter. This variability of parameter contributions to groundwater flux sensitivity is consistent with the amount of available water in the system across diverse seasons. In Summer, when soils are generally dry, small




variations in the amount of water reaching the soils surface can trigger threshold effects that influence
the amount of water transpired and evaporated and, therefore, the availability of water for groundwater
recharge. Otherwise, in Winter, when soils are often quite wet, the rate of root zone drainage can have
a major impact on the amount of water recharging the aquifer. During the latter period, a small degree
of sensitivity is also recorded for the root zone layer thickness.

To summarize the key results of the sensitivity analysis conducted for the evaporation, transpiration,
and groundwater recharge rates Table 5 lists for each model output the major sensitive parameters,
identified by a '✓' sign.

| | Albedo | Vegetation | | | | Litter | | | Root | | |
|---|---|---|---|---|---|---|---|---|---|---|---|
| | | $LAI$ | $\kappa_p$ | $K_{ext}$ | $g_s^{max}$ | $\theta_c$ | $T_L$ | $\kappa_d$ | $\theta_c$ | $T_L$ | $\kappa_d$ |
| **Evaporation** | | ✓ | | ✓ | | ✓ | ✓ | ✓ | ✓ | | |
| **Transpiration** | ✓ | ✓ | | ✓ | ✓ | | | | | | |
| **Groundwater recharge** | | | ✓ | | | | | | | ✓ | ✓ |

Table 5. Sensitivity of the target outputs of NIHM-LSM to uncertain input parameters ($\kappa_p$: rainfall
interception; $K_{ext}$: radiation attenuation; $g_s^{max}$: maximum stomatal conductance; $\theta_c$: field capacity; $T_L$:
layer thickness; $\kappa_d$: drainage rate). Sensitive parameters are identified by a '✓' sign.

Most of the results summarized in Table 5 are intuitive, groundwater recharge being an exception.
Evaporation only occurs in the top litter layer and is directly related to the energy flow through the
canopy that then reaches ground surface. As one could expect, transpiration appears as mainly
influenced by the vegetation characteristics and by albedo that influences the incoming radiation
Surprisingly, groundwater recharge is not sensible to any vegetation parameters, except the root layer
thickness and the intercept. This is possibly related to the observation that groundwater recharge





appears only when precipitations are significant and/or when transpiration rates are very small due to
a reduced energy (for example during Winter). When transpiration is significant, recharge to
groundwater takes place solely after a period of precipitations that allows transpiration and
replenishment of the water stored in the unsaturated zone.

**4 Conclusion**

We focus on the diagnosis of the behavior of the recently developed NIHM (Normally Integrated
Hydrological Model) modular Land Surface model. The latter embeds a variety of critical hydrological
processes and, similar to other land surface models, is characterized by a marked degree of
parametrization. Temporal dynamics of water fluxes associated with transpiration, evaporation, and
groundwater recharge are analyzed through global sensitivity analysis to discriminate the relative
importance of uncertain model parameters. Uncertainty sources comprise incomplete knowledge of
monthly values of albedo and leaf area index, as well as of parameters related to vegetation and soil
types constituting the litter layer and root zone. As opposed to previous studies on sensitivity analyses
of land surface models, we provide an assessment of various aspects of sensitivity upon considering a
joint analysis of multiple GSA metrics. These enable us to quantify the relative importance of our
knowledge of a given model parameter on sensitivity metrics associated with the whole probability
distribution (Eq. 3) or the first two statistical moments (i.e., mean and variance; Eqs. 4, 6 and 7) of the
density function of the target model outputs. Our analyses are exemplified through the simulation of
realistic field settings characterizing two watersheds in the Vosges region (France) across a one-year
period. Our study leads to the following major conclusions.
1. The strength of the relative importance of model parameters typically varies in time and
depends on the statistical moment associated with the probability distribution of the model
output of interest. For example, we document that the relative influence of parameters related
to the litter layer is generally higher for the variance than for the expected value of evaporation





in both catchments analyzed (Fig.s 6 and 7). The attenuation coefficient mainly affects the
expected value of transpiration as compared to its variance (Fig. 8), the relative importance of
stomatal conductance being more marked for the variance than for the average.
2.  Water fluxes related to evaporation are chiefly influenced by the energy flow through the
canopy and by the parameters characterizing the top litter layer. Transpiration appears to be
mainly influenced by the vegetation characteristics and by albedo rather than by soil-related
parameters, which play a very minor role. Groundwater recharge is influenced only by a very
limited number of model parameters. Our result document that its mean and variance are
mainly driven by the soil-related parameters, root layer thickness and intercept, while
uncertainty in the remaining vegetation parameters is somehow unexpectedly not contributing
to these. While most of these results can be intuitive, resting on rigorous GSA metrics yields
an appropriate quantification of the relative strength of the way uncertainties related to model
parameters propagates onto different statistical moments of the probability distribution of the
modeled water fluxes. Since characteristics of the soils related to the litter layer and root zone
play an important role in the evaluation of the evaporation and groundwater recharge fluxes,
our results emphasize the need for targeted studies on modeling of flow across the soil
component to best characterize these model outputs. Otherwise, the evaluation of these water
fluxes would require a priori values for some vegetation-related parameters such as canopy
height and stomatal conductance.
3.  Relying on multiple sensitivity metrics, each focused on a given aspect of the uncertainty
associated with a model response of interest, contributes to enhance our ability to quantify the
relative importance of uncertainties linked to parameters of multiple origins. While a moment-
independent analysis of the type linked to the distribution-based Borgonovo index may be
subject to some operational constraints because of the need of assessing the complete
probability density function of the model outcome of interest, it can nevertheless be employed
as a measure of the overall impact of a model parameter on the probability distribution of the





water fluxes considered. When coupled with prior knowledge of the system functioning (as for
example in the case where some parameters are not involved in the computation of the water
flux of interest), the results associated with this metric can be employed to gauge the quality of
sampling of the model parameter space (see Section 3.2). The total Sobol indices and the *AMAV*
indices provide very similar results in terms of ranking parameter importance with respect to
water fluxes variance.
We recall that each land surface model implements various degrees of complexities for diverse
processes. It is also recognized that uncertainty sources affecting land surface models typically
comprise incomplete knowledge in (*a*) conceptual and mathematical formulation of models and
processes therein and (*b*) parameters embedded in such models. As such, future research efforts will
be aimed at extending our knowledge on the relative impact of uncertain processes (and their
parameterization) on the different components of the water budget included in a land surface model.




**Code and data availability:**

The code and data will be made available using a shared folder upon request.

**Author contribution:**

PA, AD, AG and SW designed the research; DL produced the data; all authors contributed to the analysis and interpretation of the results; DL and PA produced figures, tables and the first draft. All authors contributed to improve the manuscript. AG and PA wrote the submitted versions, using feedback from all the co-authors.

**Competing interests:**

The contact author has declared that none of the authors has any competing interests.
Some authors are members of the editorial board of HESS.

**Acknowledgements**

Daniel Luttenauer acknowledges support of the French Ministry of Agriculture and ENGEES for support through FCPR.
Alberto Guadagnini acknowledges the *Chaire Gutenberg* (Région Grand Est France, Strasbourg City and Strasbourg University).
Philippe Ackerer acknowledges support of the Politecnico di Milano (grant Senior Resident Researcher 2023).
Aronne Dell'Oca acknowledges support of the National Recovery and Resilience Plan (NRRP), Mission 4 Component 2 Investment 1.4 - Call for tender No. 3138 of December 16, 2021, rectified by Decree n.3175 of December 18, 2021 of Italian Ministry of University and Research funded by the European Union – NextGenerationEU; Project code CN_00000033, Concession Decree No. 1034 of June 17, 2022 adopted by the Italian Ministry of University and Research, CUP D43C22001250001, Project title "National Biodiversity Future Center - NBFC".





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
