# Peer review of "Relative importance of uncertain model parameters driving water fluxes in a Land Surface 1 2 Model 3 4 5 6 David Luttenauer1, Aronne Dell'Oca2, Alberto Guadagnini2, Sylvain Weill1, Philippe Ackerer1 7 1 Institut Terre et Environn"

_Hydrology and Earth System Sciences, 2024_

## Author Comment (AC1)

**Reviewer #2**

This paper by Lutternauer et al. performed parameter sensitivity analysis of a land surface model at two watersheds in France. The authors employ various sensitivity analysis methods to assess an extensive set of model parameters, which is quite interesting.

We thank the Reviewer for the encouraging assessment of our work. We provide in the following our answers to the comments emerged.

However, I do have some concerns that I feel need to be addressed:

1. My major concern is on the water balance at those two catchments. The average evapotranspiration is only 38.6% and 11.6% of total precipitation in those two catchments, which means runoff (surface and subsurface) must be large. Does that align with the observations at those catchments? Are there observations available to evaluate the model performance? I am concerned because I feel that the sensitivity analysis is most useful when the parameter values are sampled around their optimal values in the multi-dimensional parameter space. Otherwise, the analysis may not reflect the real parameter sensitivity. For example, in an extreme case, if the model simulates predominantly surface runoff with minimal evapotranspiration, parameters linked to evapotranspiration would exhibit weak sensitivity, which does not reflect reality. It would be helpful if the authors can show the observations of discharge, or evapotranspiration, if available.

With reference to the evaluation of model performance, we stress that our purpose is not to perform model calibration. Rather, we are placing our study in the context of model diagnosis. Hence, we place our study in the context of *ab initio* global sensitivity analysis. As we state in the original manuscript, this is a critical step that needs to be performed prior to model calibration. It provides insights on model functioning and on the way its response is affected by parameter uncertainty. In this sense, our global sensitivity analysis is performed prior to model calibration. One can then perform a global sensitivity analysis after model calibration and assess the way residual parameter uncertainty (i.e., conditional on available data) influences the residual uncertainty associated with model outputs. This second step has a different purpose than the one we consider in our study, as recognized in a variety of studies.

Prompted by the Reviewer's comment, we will place further emphasis on this point in our revised Introduction.

2. Related to the first point, I feel the paper can be strengthened if all components of the water balance can be included.

As we state in our answer to the first comment of Reviewer #1, we did not take surface runoff under consideration. This is tied to the fact that it was not observed at either of the two sites. The soil is quite permeable in both catchments. It is mostly covered by vegetation and rain intensity is not high enough to generate significant runoff under those conditions. As such, our conclusions hold for sites or situations where runoff can be considered as negligible when compared to the other water flux component (evaporation, transpiration, and groundwater recharge). This aspect will be clarified in the revised manuscript.

3. The authors mentioned that the model was run in a distributed way, but only selected one grid for each catchment for analysis. I am wondering if the model is three-dimensional. From what I read, the model seems to be a one-dimensional grid model. If it is one-dimensional, running the other grids should not affect the results. Some clarification would be very helpful.

The hydrological model NIHM is a two-dimensional (in the horizontal plane) physically based coupled surface/subsurface model. For each element/cell of the 2D mesh, the recharge is computed by the land surface model NIHM-MLSM (the detailed description of the model is provided in the Supplementary Material). The Global Sensitivity Analysis concerns NIHM-MLSM. As we clarify in the Supplementary Material and in the body of the manuscript, the latter is a one-dimensional model which has to be run for each element/cell of the mesh associated with NIHM.

Prompted by the Reviewer's comment, we will add some additional details about this point in the revised manuscript.

4. The paper does not have a "discussion" section, which limits the paper's impact. I feel some discussion would strengthen the paper substantially. For example, how does the sensitivity analysis results compare with other studies? How do the four sensitivity analysis methods differ? Do the sensitivity analysis results reveal some important insights into the model mechanisms, or the hydrological conditions at those catchments?

Prompted by the Reviewer's comment, we will: (*a*) expand on the meaning of the sensitivity indices employed; and (*b*) reorganize the session on results.

As we state in our reply to Reviewer #1, the results of a (global or local) sensitivity analysis are model dependent. Even considering diverse LSMs sharing some parameters, the relative weight of some parameters may change. Otherwise, we think that, since water availability is a key variable for evaporation, transpiration and groundwater recharge, soil related parameters will play an important role. We will explicitly address this issue in the revised manuscript.

5. I feel it could be helpful to show the values of LAI and albedo in the manuscript, either using a figure or a table.

The following table will be provided in the revised manuscript. It lists monthly averaged values for Albedo and LAI.

| | Month | | | | | | | | | | | |
|---|---|---|---|---|---|---|---|---|---|---|---|---|
| | **1** | **2** | **3** | **4** | **5** | **6** | **7** | **8** | **9** | **10** | **11** | **12** |
| **Bruche** | | | | | | | | | | | | |
| **Albedo** | 0.208 | 0.155 | 0.137 | 0.136 | 0.168 | 0.190 | 0.177 | 0.158 | 0.164 | 0.179 | 0.202 | 0.543 |
| **LAI** | 1.101 | 0.869 | 1.316 | 2.120 | 3.959 | 4.392 | 4.633 | 4.401 | 4.333 | 3.540 | 1.212 | 0.850 |
| | | | | | | | | | | | | |
| **Doller** | | | | | | | | | | | | |
| **Albedo** | 0.187 | 0.152 | 0.143 | 0.149 | 0.169 | 0.195 | 0.179 | 0.167 | 0.173 | 0.176 | 0.190 | 0.268 |
| **LAI** | 1.091 | 0.809 | 1.229 | 2.147 | 4.264 | 4.79 | 4.906 | 4.757 | 3.730 | 3.730 | 1.043 | 0.867 |

Table XX: Monthly averaged values for Albedo and LAI for the two locations considered in the Global Sensitivity Analysis.

**Specific comments:**

1. Units of field capacity and porosity in Table 2 are missing.

They were not included because these are dimensionless quantities. We will make this clear in the revised manuscript.

2. *Line 540: " For example, the value of B for the evaporation at Bruche during the month of July associated with the LAI of January must be zero. However, inspection of Fig. 6a does not reflect this anticipated outcome. This apparent anomaly is attributed to a random noise…".* I don't quite agree with the authors. The results from previous steps might affect future steps. I don't think the B value for July evaporation associated with January LAI is necessarily zero.

We agree that this assumption can be questioned. In the revised manuscript, we will address this point considering the impact of the root zone related parameters on evaporation because these parameters are not involved in the computation of the evaporation (evaporation is assumed to occur in the litter zone only).

3. Figures 6 to 11 are difficult to read. Can parameter symbols be used instead of parameter index in those figures? Or at least, put the parameter identification codes in the manuscript, instead of the supplemental materials?

We will address this point in the revised manuscript.

4. What are $\theta_L$ and $\theta_p$ in Equation 25 in the supplementary material?

We apologize for the typos in the equation, that is now being rectified.

5. Can authors add how surface runoff is determined? I think that could help the readers understand some of the results.

As we state in our answer to the first comment of Reviewer #1 and comment #2 of this Reviewer, we did not take surface runoff under consideration, given the conditions of the two catchments considered.

---

## Author Comment (AC2)

Reviewer #1

**GENERAL COMMENT:**

This paper shows the results of a comprehensive sensitivity analysis for three key hydrological variables (evaporation, transpiration and recharge) using a complex land surface model, the NIHM modular LSM. The sensitivity analysis is based on three metrics that evaluate the relative weight of different model parameters on the probability distribution, expected value and variance of the chosen variables. In addition, an interesting feature of the analysis is to identify the temporal change in relative weight of the parameters. On the other hand, although the authors choose two different locations to perform the sensitivity analysis, the spatial dimension seems less important and does not seem to be addressed by the results.

I believe that the work of Luttenauer et al. shows interesting results that could help to propose improvements in LSMs with respect to parameter values. It could help to open new perspectives to correct some flaws that could be observed when evaluating an LSM against observed data. But also, I think the article could be improved, especially in the discussion part. In my opinion, the authors address the problems they point out in the introduction, i.e., uncertainty in the knowledge of parameter values, and the need to identify important parameters, but they do not present the limitations and differences of their results compared to other examples in the literature. Following the same line, the authors do not attempt to state some general conclusions that could be useful for other models and scales, although I understand the difficulties in reaching such conclusions. In conclusion, I think the manuscript could benefit from showing some additional results, so that the results are better understood, and from a longer discussion, so that the implications and perspectives of these results are clearly stated and compared with other efforts in the literature.

We thank the Reviewer for the encouraging assessment of our work. We provide in the following our answers to the comments emerged.

**MAJOR COMMENTS**

I would like to begin with two observations regarding the results presented.

First, LSMs differ from classical hydrological models mainly because they solve water and energy balances at the same time, estimating actual evapotranspiration fluxes as a function of available energy and water. Here I understand that the focus of the paper is water fluxes for a single pixel, so I will not consider any other energy-related variables, but I believe that at least one variable is missing in the analysis: surface runoff. Including surface runoff in Tables 3 and 4, and Figures 3 and 4 should be sufficient to better understand the response of the water system.

We did not take surface runoff under consideration because it was not observed on both sites. The soil is quite permeable in both catchments. It is mostly covered by vegetation and rain intensity is not high enough to generate significant runoff under those conditions. As such, our conclusions hold for sites or situations where runoff can be considered as negligible when

compared to the other water flux component (i.e., evaporation, transpiration, and groundwater recharge). This aspect will be clarified in the revised manuscript.

Also, if possible, I think the authors should include in the supplement a figure with the delimitation of the two river basins, and the pixels that were included in the analysis.

We will provide this figure in our revised manuscript.

The second observation concerns the relationship between LAI and albedo. The authors say that both parameters come from remotely sensed data, but also that this global albedo is split into soil and canopy albedos (section 2 of the supplement). Does this mean that albedo participates in the estimation of net radiation, but that both albedo and LAI participate in the partitioning of evapotranspiration between evaporation and transpiration? If so, that might explain why evaporation (which, I assume, always occurs under the canopy layer, i.e., the pixel does not have an area with bare soil fraction) is not sensitive to albedo values. If so, I would recommend the authors to clarify this relationship in section 2 (specifically line 333), and to consider it in the results section. Also, I'm not sure if I missed this, but there is no value for bare soil albedo (in line 190 of the supplement, it is assumed that the soil albedo is known). Please put the value used in the supplement.

Evaporation occurs under the canopy layer and depends on the amount of energy reaching the soil surface (i.e., the net radiation). Following Taconet et al. (1986), we consider the canopy layer as a semi transmissive layer and compute net radiation at the canopy and the soil surface using equation (1) and (2). The LAI appears in both equations and albedo values for the canopy and the soil are required. Satellite data provide an albedo value at the pixel scale and the canopy albedo is computed assuming the soil albedo value as known (see equation (28) in the supplementary material). Notice that this estimate of the net radiation is computed even if the pixel does not have a bare soil fraction and that the soil albedo is the albedo of a vegetated area. We set it to a value of 0.20 (Bonon, 2008).

We agree that this point is not well addressed in our manuscript. We will clarify this in the revised manuscript and supplementary material.

**References**

Taconet, O., Bernard, R., and Vidal-Madjar, D., Evapotranspiration over an Agricultural Region Using a Surface Flux/Temperature Model Based on NOAA-AVHRR Data, J. Clim. Appl. Meteorol., 25, 284-307, https://doi.org/10.1175/1520-0450(1986)025<0284:EOAARU>2.0.CO;2, 1986.

Bonan, G., Forests and Climate Change: Forcings, Feedbacks, and the Climate Benefits of Forests, Science 320, 5882, 1444-49. https://doi.org/10.1126/science.1155121, 2008.

As for the conclusions, I have two main observations:

First, I believe that the authors have not sufficiently related the results to the climatic and local conditions of the two basins. For example, with respect to groundwater recharge, and its sensitivity mainly to soil-related parameters, one would think that the two basins have sufficient water for evapotranspiration (i.e., the basins are energy-limited), and therefore, vegetation does not play an important role in the recharge rate.

We will improve the description of climatic conditions by including details about potential evaporation and transpiration. In the two settings considered, there is not enough stored water in the soil layers to satisfy the demand for evaporation and transpiration. In such scenarios, independent of the values of the vegetation related parameters that govern evaporation and transpiration, groundwater recharge is very limited as long as the available water quantity stored in the litter and root layers is smaller than potential evaporation and transpiration.

This would also mean that the partitioning of runoff between surface runoff and baseflow is also controlled by soil parameters. This idea seems consistent with the precipitation rates in both catchments (903 and 2541 mm) and with a small sensitivity of recharge to LAI values between May and October. Also, with a slightly higher sensitivity in the Bruche basin for LAI.

As we state above, the water demand to satisfy evaporation and transpiration is higher than the available water quantity stored in the litter (for evaporation) and in the root (for transpiration) layers. Therefore, the LAI values do not impact the simulated groundwater recharge.

Prompted by the Reviewer's comments, we will improve the discussion related to sensitivity of groundwater recharge to vegetation related parameters.

Second, I think the manuscript needs an effort to further discuss the implications of these results. For example, what are the implications of the sensitivity analysis for other land surface models? Is it possible for other land surface models (such as those used in climate models) to guide a parameter calibration based on these results? Or is it still acceptable to use simplified, a priori parameter values?

An *ab initio* uncertainty analyses should always be performed prior to running a land surface model. Doing so enables one to identify parameters and processes that require special attention due to their impact on model results. It will also guide model calibration because un-sensitive parameters cannot be calibrated and an a priori value can be acceptable. With reference to model calibration, we recall that a parameter can be sensitive while not being identifiable because of its strong correlation to other parameters. For example, if we consider the model $y = (a + b) x$ (where $y$ is the model output, $x$ is model state variable and (a, b) are model parameters), $y$ is sensitive to both parameters and a and b cannot be calibrated individually. We will include a short discussion about these elements in the revised manuscript.

What are the implications of these results for a model that includes new processes, which could change the relative weight of some parameters?

The results of a (global or local) sensitivity analysis are of course model dependent. Even considering diverse LSMs sharing some parameters, the relative weight of parameters may change. Otherwise, we think that, since water availability is a key variable for evaporation, transpiration and groundwater recharge, soil related parameters will play an important role. We will explicitly address this issue in the revised manuscript.

How could we address the complexification of LSMs and introduction of new features and processes, at least at the local scale used here?

This is precisely the benefit one can obtain upon relying on an *ab initio* global sensitivity analysis. The latter will yield quantitative appraisal of the importance of new features/processes that can eventually arise when considering various levels of complexity for a given model formulation. We have analyzed this specific aspect in a recent study (Ceresa et al., 2023; already referenced in the original manuscript). Albeit such study is focused on groundwater contamination by a given pharmaceutical product, it provides guidelines about the way global sensitivity analysis can be employed to explore the relative importance of diverse processes embedded in an interpretive model. We will include a short comment about this element in the revised manuscript.

**References**

Ceresa, L., Guadagnini, A., Rodríguez-Escales, P., Riva, M., Sanchez-Vila, X., and Porta, G. M., On Multi-Model assessment of complex degradation paths: The fate of diclofenac and its transformation products. Water Resources Research, 59, e2022WR033183. https://doi.org/10.1029/2022WR033183, 2023.

I understand that this LSM is run at a local scale, while other LSMs are run at regional and global scales, so there is a scale issue regarding this analysis. Also, that there are important differences in model structure and processes representation that could prevent to state general conclusions. But the authors could use the results to further discuss the prospects of defining parameter values more carefully, and using sensitivity analysis to reduce the uncertainty of key variables for complex models such as LSMs.

We agree and will strengthen the Conclusions upon addressing the concept of scales and reliance on LSMs characterized by increased complexity levels.

I add some references that may be interesting for this discussion. For parameter sensitivity at different scales:

(https://hess.copernicus.org/articles/24/3753/2020/);
(https://onlinelibrary.wiley.com/doi/10.1029/2019WR026612);

An example of changes in the model output due to new processes or differences in the mathematical representation:

(https://gmd.copernicus.org/articles/17/2141/2024/);
(http://doi.wiley.com/10.1002/2016JD025426)

We are grateful to the Reviewer for pointing us to these additional references and will include those relevant to our work in the revised manuscript. These include the work by Tafasca et al. (2020; https://hess.copernicus.org/articles/24/3753/2020/), that shows that soil physical properties play an important role in estimating soil water and energy fluxes, as well as the study by Maina et al. (2020; https://onlinelibrary.wiley.com/doi/10.1029/2019WR026612), who rely on a semisynthetic test setting and perform a global sensitivity analysis based on Sobol and AMAE indices through a surrogate model (constructed through a polynomial chaos expansion approximation) to assess impacts of subsurface physical properties on evapotransporation.

Otherwise, the study by Arboleda-Obando et al. (2024; https://doi.org/10.5194/gmd-17-2141-2024) illustrates a new irrigation scheme upon relying on the land surface model ORCHIDEE (ORganising Carbon and Hydrology in Dynamic EcosystEms)) and is focused on conditioning actual irrigation according to available freshwater by including an environmental threshold and allocation rules that are related to local infrastructure. Finally, the work by Keune et al. (2016; http://doi.wiley.com/10.1002/2016JD025426) is chiefly focusing on groundwater and its impact (as evaluated through a scatterplot and an ANOVA approach) on feedbacks on land-surface and atmosphere during a specific event, i.e., the European heat wave that was observed in 2023. In this sense, we do see these as only marginally related to our work and would prefer to avoid referencing them. We will naturally abide by the Editor's decision on this point.

**References**

Tafasca, S., Ducharne, A., Valentin, C., 2020. Weak sensitivity of the terrestrial water budget to global soil texture maps in the ORCHIDEE land surface model. Hydrol. Earth Syst. Sci. 24, 3753–3774. https://doi.org/10.5194/hess-24-3753-2020

Maina, F.Z., Siirila-Woodburn, E.R., 2020. The Role of Subsurface Flow on Evapotranspiration: A Global Sensitivity Analysis. Water Resources Research 56, e2019WR026612. https://doi.org/10.1029/2019WR026612

Arboleda-Obando, P. F., Ducharne, A., Yin, Z., and Ciais, P.: Validation of a new global irrigation scheme in the land surface model ORCHIDEE v2.2, Geosci. Model Dev., 17, 2141–2164, https://doi.org/10.5194/gmd-17-2141-2024, 2024

Keune, J., F. Gasper, K. Goergen, A. Hense, P. Shrestha, M. Sulis, and S. Kollet (2016), Studying the influence of groundwater representations on land surface-atmosphere feedbacks during the European heat wave in 2003, J. Geophys. Res. Atmos., 121, 13,301-13,325, doi:10.1002/2016JD025426.

Finally, I would suggest describing the mode setting in more detail. It is not clear if the model uses 8 km resolution like Safran, or if it is finer. Also, how does the model deal with land surface heterogeneity? On line 383 the authors say that only one vegetation type is used for Bruche, and two vegetation types for Doller. If the model uses an 8 km pixel, what is the implication of simplifying land surface heterogeneity for sensitivity analysis (if there is any implication)?

The size of the *computational pixel* (i.e., the size of the pixel at which energy and water balance are computed) is not provided in the original manuscript. It is around $200 \times 200$ m$^2$. We do apologize for this oversight. Section 2.3 will be slightly modified to provide additional information about scales.

**MINOR OBSERVATIONS**

Some additional observations and suggestions:

Decharme et al., 2019 is not in the references section. On the other hand, Decharme et al., 2011 is in references but not in the text. They both refer to ISBA model, Decharme et al., 2011 is interested in the sensitivity of the model to pedo-transfer functions in a soil multilayer representation, and Decharme et al., 2019 presents the ISBA-CTRIP model. Please review the reference list, and clarify the reference within the text.

We will thoroughly check the consistency between the main text and the reference list.

Also, here the authors are putting together regional and global scale LSMs. While essentially the same, regional models could represent in more details some processes, like horizontal groundwater flow, while global models try to parameterize the main processes in an idealized way. Two additional regional models: Catchment (Koster et al., 2000), LEAF2 (Fan and Miguez-Macho 2011), and one additional global model: ORCHIDEE (Krinner et al., 2005).

We will consider this element in the structure of the Introduction.

128: designed

132: "(a) 133 application of a unique model formulation across different soil and vegetation types is questionable" this phrase may be reformulated

334: I think that here, when you say "time" you refer to "moment"

384: not sure the word "exemplary" must be used here

519: It is difficult to say because curves are close between them, but it seems that unconditioned PDF and PDFs for higher albedo values are similar, so to say that higher transpiration values are controlled by higher albedo values is not straightforward for me.

Fig. 6 As Sobol and AMAV indices are similar (they both are based on variance), I would suggest to put them in the same column

621: maybe just put "by the drainage from the litter layer"

For supplementary, please numerate and put a caption to tables in section 3. Also, water content at saturation uses in line 156, and in table in section 3.

We will consider in details all of these comments and suggestions in our revised manuscript.